**Investigation**

# High-throughput genetic manipulation of multicellular organisms using a machine-vision guided embryonic microinjection robot

Andrew D. Alegria,[1,‡] Amey S. Joshi,[1,‡] Jorge Blanco Mendana,[2] Kanav Khosla,[1] Kieran T. Smith,[3] Benjamin Auch,[2] Margaret Donovan,[2] John Bischof,[1,4] Daryl M. Gohl (iD) ,[2,5,*] Suhasa B. Kodandaramaiah (iD) [1,6,*]

[1]Department of Mechanical Engineering, University of Minnesota, Minneapolis, MN 55455, USA
[2]University of Minnesota Genomics Center, University of Minnesota, Minneapolis, MN 55455, USA
[3]Department of Fisheries, Wildlife and Conservation Biology, University of Minnesota, St. Paul, MN 55108, USA
[4]Department of Biomedical Engineering, University of Minnesota, Minneapolis, MN 55455, USA
[5]Department of Genetics, Cell Biology and Development, University of Minnesota, Minneapolis, MN 55455, USA
[6]Department of Neuroscience, University of Minnesota, Minneapolis, MN 55455, USA

*Corresponding author: 2231 6th Street SE, Minneapolis, MN 55455, USA. Email: dmgohl@umn.edu; *Corresponding author: Mechanical Engineering, 111 Church St SE, Minneapolis, MN 55455, USA. Email: suhasabk@umn.edu
‡These authors contributed equally.

Microinjection is a technique used for transgenesis, mutagenesis, cell labeling, cryopreservation, and in vitro fertilization in multiple single and multicellular organisms. Microinjection requires specialized skills and involves rate-limiting and labor-intensive preparatory steps. Here, we constructed a machine-vision guided generalized robot that fully automates the process of microinjection in fruit fly (*Drosophila melanogaster*) and zebrafish (*Danio rerio*) embryos. The robot uses machine learning models trained to detect embryos in images of agar plates and identify specific anatomical locations within each embryo in 3D space using dual view microscopes. The robot then serially performs a microinjection in each detected embryo. We constructed and used three such robots to automatically microinject tens of thousands of Drosophila and zebrafish embryos. We systematically optimized robotic microinjection for each species and performed routine transgenesis with proficiency comparable to highly skilled human practitioners while achieving up to 4× increases in microinjection throughput in Drosophila. The robot was utilized to microinject pools of over 20,000 uniquely barcoded plasmids into 1,713 embryos in 2 days to rapidly generate more than 400 unique transgenic Drosophila lines. This experiment enabled a novel measurement of the number of independent germline integration events per successfully injected embryo. Finally, we showed that robotic microinjection of cryoprotective agents in zebrafish embryos significantly improves vitrification rates and survival of cryopreserved embryos post-thaw as compared to manual microinjection. We anticipate that the robot can be used to carry out microinjection for genome-wide manipulation and cryopreservation at scale in a wide range of organisms.

Keywords: robotics; transgenesis; computer vision; mutagenesis; screening

## Introduction

The microinjection of microscopic objects such as cells or embryos is an important technique used in biomedical research as it enables a wide range of genetic applications including transgenesis, targeted mutagenesis, perturbation via small molecules or dsRNA, labeling or monitoring via dyes, transplantation, cloning, and in vitro fertilization (Zhang and Yu 2008; Shull *et al.* 2019; Shull *et al.* 2021).

Protocols for successful microinjection and transgenesis have been crucial for the continued utility and prominence of model organisms such as mice, zebrafish (*Danio rerio*), *Caenorhabditis elegans*, and Drosophila. Injection-based methods are used for transposon-mediated transgenesis, targeted integration (via PhiC31 or other recombinases), as well as CRISPR/Cas9 gene editing or dsRNA-mediated gene silencing, and to introduce other components such as labeled proteins for studying cellular

processes. In turn, these methods have enabled a slew of sophisticated genetic techniques for targeted gene expression and manipulation, studying development and gene function, and the nervous system (Janik *et al.* 2000; Rosen *et al.* 2009; Del Valle Rodríguez *et al.* 2011; Venken *et al.* 2011; Schubert *et al.* 2014; Nance and Frøkjær-Jensen 2019; Abdelrahman *et al.* 2021). In addition to genetic manipulation, microinjection is also used for other applications such as cryopreservation where it is one of the most effective methods for introducing cryoprotective agents into the yolk of the zebrafish embryo (Janik *et al.* 2000). Microinjection can also be used to prepare model organisms such as zebrafish for cryopreservation (Khosla *et al.* 2017, 2020; Guo *et al.* 2024) or possibly Drosophila or other invertebrates with large relatively impermeable embryos in the future (Zhan *et al.* 2021).

The microinjection procedure—i.e. a brief insertion of the microneedle into a specific anatomical or subcellular location under pressure to eject a controlled volume of microinjectant is

essentially identical across targeted cells or microorganisms. However, microinjection into each specific target is a specialized skill that often requires significant practice to attain proficiency. Manual microinjection protocols further suffer from low throughput. Successful microinjection is thus highly operator-dependent and limits the application of the technique to labs with the resources to support staff with this specialized skill. In addition, biological variables can impart constraints on the timing or location of microinjection. For example, in Drosophila, the germ cells form in the posterior pole of the embryo within approximately 90 min of fertilization (Bownes 1975). Thus, to modify the germline with microinjected material, the injected payload must be delivered to the posterior pole within 1–1.5 h of egg laying to be encapsulated in the germ cells. Similarly, microinjection sessions with zebrafish embryos must be limited to 45 min to prevent desiccation (Janik *et al.* 2000).

The recent development of new tools for large-scale synthesis and sequencing of DNA, coupled with advances in genome editing have made microinjection a critical bottleneck in conducting large-scale experiments in animal models. While several large-scale microinjection-based transgene collections have been established, these efforts have involved a significant expenditure of resources (Dietzl *et al.* 2007; Manning *et al.* 2012). Although some efforts have been made to automate the microinjection of Drosophila or zebrafish embryos (Zappe *et al.* 2006; Wang *et al.* 2007; Cornell *et al.* 2008; Delubac *et al.* 2012; Ghaemi *et al.* 2017), these systems have not been widely adopted due to their bespoke hardware and software and lack of transformative improvements in speed and scale. The development of robust methods for large-scale automated microinjection has the potential to transform the scope and scale of genetic manipulation and enable new types of experiments and applications when paired with other omics technologies.

We have developed a versatile, generalized robotic microinjection platform that targets embryos directly on agar plates, minimizing specimen handling and manual operations to increase throughput and minimize reliance on operator skills. Using machine learning (ML) models, we show that embryos and microinjection points can be reliably identified and targeted and that microinjection parameters can be quickly assessed and optimized. We demonstrate that the automated microinjection robot can microinject both Drosophila and zebrafish embryos to carry out transposon-mediated transgenesis, site-specific integration, and CRISPR/Cas9 mutagenesis at success rates comparable to or exceeding manual microinjection, and at speeds greater than highly trained manual microinjection practitioners. Finally, we show that the automated microinjection robot has the potential to unlock new microinjection-based applications such as whole zebrafish cryopreservation both by enabling high throughput and scalable microinjection workflows and by improving the precision of microinjection.

## Methods
### Robot construction
The microinjection robot uses an XYZ stage, consisting of three DC motors with rotary encoders for sensor signals, which are controlled using a proportional, integral, derivative (PID) controller. A custom-designed plate holder is mounted on the XYZ stage. The plate holder incorporates a circular light-emitting diode (LED) illuminator which is used to uniformly illuminate the agar plate during robot operation. Inclined microscopes are custom-built using a complementary metal oxide semiconductor (CMOS) microscope camera and a 2× magnification objective lens (Pierce *et al.* 2011). Both the inclined

microscopes are positioned such that they can simultaneously image a fixed micropipette and embryo on the agar plate from two different perspectives. The two inclined microscopes allow the robot to estimate the embryos' microinjection point and micropipette tip location in 3D space. For the manufacturing of the micropipettes, we used glass aluminosilicate capillaries with filament and pulled the capillaries using a Sutter P-1000 micropipette puller. The micropipettes were then beveled using a Sutter BV-10 micropipette beveler to an ~3 µm micropipette tip opening. Two methods were developed to deliver nanoliters of microinjectant once the micropipette is inside the embryo. These two methods work on the principle of positive displacement, and hydrostatic pressure. For the positive displacement method, a small metal plunger, which was finely controlled using a stepper motor controller (Nanoliter 2020, Injector (300704), World Precision Instruments), was used to deliver the microinjectant. For the hydrostatic pressure method, air pressure was applied at the back of the micropipette, which was controlled using an electronic pressure regulator (QPV Series Proportional Pressure Regulator, Proportion-Air Inc) and a microcontroller (Arduino Nano), to deliver the microinjectant. All the hardware is connected to the computer via digital link connections.

A graphical user interface (GUI) was developed to facilitate use of the automated microinjection system. Object detection is the rate-limiting step for injection and needs to be executed using a Graphics processing unit (GPU). To improve the speed of the overall process, object detection using the ML models was run using a separate thread in the python code (Supplementary Fig. 1). The output of each object detection, that is microinjection point pixel coordinates were the only shared memory resources between the threads.

## Training ML models
The microinjection robot relies upon fast and accurate detection and classification of embryos, micropipette tips, and location within embryos. In addition, object recognition needs to be performed near real-time (15–20 Hz). Faster Region Convolution Neural Network (Faster R-CNN) (Ren *et al.* 2017), was used to detect different classes of objects for Drosophila microinjection (e.g. single embryos, micropipettes, micropipette tips, and landmarks/positions within the Drosophila embryos). While, You Only Look Once (YOLO) v4 (Bochkovskiy *et al.* 2020) was used for zebrafish embryos for detection (e.g. single and dead zebrafish embryos, yolk, micropipette tip, microinjection point).

### *Data collection and preparation for model training*
Images were collected during the first 24 h after the collection of Drosophila and zebrafish embryos. Embryos were randomly distributed across a 9 cm agar plate and images were captured using a DSLR camera. In the case of zebrafish embryos, images were captured at different developmental stages to allow detection at any developmental stage. Images contained a combination of alive and dead zebrafish embryos, which can be distinguished by the opacity of an embryo. The data contains a total of 235 images of Drosophila embryos with $6,000 \times 4,000$ pixels resolution and 945 images of zebrafish embryos with $1,000 \times 1,000$ pixels resolution.

Similarly, for the dataset for microinjection point and micropipette tip detection, images containing zebrafish and Drosophila embryos along with the micropipette were captured using the inclined microscopes. Images contained combinations of zebrafish embryos at different developmental stages from 1-cell to bud stage. For Drosophila embryos, images contained a

combination of embryos at different orientations. Also, the training images contained a combination of different micropipettes with various tip openings, microinjectant, and lighting conditions. The training data comprised a total of 7,545 images of Drosophila embryos and micropipette tips with $1,280 \times 720$ pixels resolution and 1,484 images of zebrafish embryos with $1,296 \times 732$ pixels resolution. Bounding box labeling was done according to each class in the ML model using LabelImg (Tzutalin 2015). Each dataset was further expanded through data augmentation to improve the neural network performance and eliminate the overfitting of the dataset during the training procedure. In this case, the contrast and blurriness of the images were altered, additionally, images were rotated randomly by 90°, 180°, or 270° for data augmentation. For each image four augmented images were used. This data was further randomly split 80:20 into training and test sets.

### Implementation of data training zebrafish image training

YOLOv4 models were trained using the Darknet framework in a cloud-based environment (Google Colab Notebook, Alphabet Inc.). A GPU, (Tesla T4, Nvidia Inc.) was used for training, validation, and inference. K-mean clustering was used for calculating custom anchors. Training-time augmentation was enabled during the training period. A linear warmup policy was used for the first 1,000 iterations during training followed by a piecewise constant decay as a learning rate schedule policy for subsequent iterations (Tata et al. 2021).

### Drosophila image training

Faster R-CNN models were trained using free open-source ML libraries (TensorFlow) in an Anaconda environment within Python. A GPU (GE Force GTX, Nvidia Inc.) was used for training, validation, and inference. An initial learning rate of 0.0003 was used for the first 90,000 iterations and then a learning rate of 0.00003 was used for subsequent iterations (Yu et al. 2020) (Supplementary Fig. 2).

### ML efficiency evaluation

To evaluate the efficiency of the ML models we calculated the average precision (AP) of the model at 50% intersection of union (IoU). AP and IoU were calculated using the following formulas (Everingham et al. 2010):

TP: True positive; FP: False positive; FN: False negative

$$\text{Intersection of union (IoU)} = \frac{\text{area of overlap}}{\text{area of union}}$$

$$\text{Recall (R)} = \frac{\text{TP}}{\text{TP} + \text{FP}}$$

$$\text{Precision (P)} = \frac{\text{TP}}{\text{TP} + \text{FP}}$$

$$\text{Average precision (AP)} = \frac{1}{11} \sum_{R^i} P.R_i$$

### Injection success evaluation

The success of the microinjections was evaluated using the following formulae:

Survival rate %
$$= 100 * \frac{\text{\# larvae or surviving zebrafish embryos by Day 5}}{\text{\# Drosophila or zebrafish embryos injected}}$$

Insertion efficiency %
$$= 100 * \frac{\text{\# Transposon mediated transgenic vials}}{\text{\# fertile crosses}}$$

Integration efficiency %
$$= 100 * \frac{\text{\# PhiC31 mediated transgenic vials}}{\text{\# fertile crosses}}$$

$$\text{Mutagenesis efficiency \%} = 100 * \frac{\text{\# mutant vials}}{\text{\# fertile crosses}}$$

$$\text{Injection efficiency \%} = 100 * \frac{\text{\# transgenic or mutant vials}}{\text{\# Drosophila embryos injected}}$$

Transformation efficiency %
$$= 100 * \frac{\text{\# GFP or RFP zebrafish embryos}}{\text{\# Zebrafish embryos injected}}$$

## Drosophila microinjection

### Optimization of robotic microinjection experiments

Ranges of values were tested for each parameter, and three trials of experiments were performed at each parameter setting. The survival rate was defined as the percentage of survived larvae compared to the number of microinjected embryos. To measure the effectiveness of the volume of solution microinjected into the embryo on survival rate, a computer vision algorithm was used to calculate in real-time—the number of pixels detected in the color of the microinjectant dye to relate the volume of microinjectant ejected into the embryo.

### Constructing nos-Ca9 line

The *nos* promoter, 5′ UTR and 3′ UTR regulatory regions were amplified by PCR from the plasmid *pNos-PhiC31* (a gift from Johannes Bischof). The Cas9 coding sequence was amplified by PCR from the plasmid *vasaCas9* (a gift from Pierre Leopold). The PCR fragments were purified and inserted into the vector *pigAct88F-GFP* (Sharkey et al. 2020) through recombinational cloning (In-Fusion, Takara). Primers used for the PCR reactions (sequences for recombination are underlined):

Nos5UTR-F <u>ACGCGTACGGCGCGCC</u>AAGCTTCGACCGTTTTAACC
Nos5UTR-R <u>CCGGCCTAGGCGCGCC</u>GGCGAAAATCCGGGTCGA AA
Nos3UTR-F <u>CGCCGGCGCGCC</u>TAGGGCGAATCCAGCTCTGGAGCA
Nos3UTR-R <u>GAACATTGTCAGATCT</u>TTCCTGGCCCTTTTCGAGAA
Cas9-1-F <u>CGCCGGCGCGCC</u>TAGGGCCACCATGGACAAGAAGTA CTCC
Cas9-1-R <u>CTGGATTCGCCCTAGG</u>TCACACCTTCCTCTTCTTCT

The final vector was injected into Drosophila wild-type embryos following standard *piggyBac* transformation protocols.

### Transgenesis and mutagenesis experiments

For piggyBac transgenesis, a plasmid with Pbac{3xP3-EGFP} was injected together with the piggyBac helper plasmid into *w-; +; +*

flies (Horn *et al.* 2000). *PhiC31*-mediated targeted transgenesis was performed in the *Drosophila* strain *P{y[ + t7.7] = nanos-phiC31 \int.NLS}X, y[1] sc[1] v[1] sev[21]; P{y[ + t7.7] = CaryP}attP2* (BDSC #25710) using the vector *pActEHG-attB*, a modified version of *pigAct88F-GFP*. For CRISPR mutagenesis, *nos-cas9* embryos were injected with 250 ng/μL of the construct *pCFD5-white-gRNA*. This construct contains the DNA sequence for a *white*-specific gRNA downstream of the *Drosophila* U6-3 promoter. The primers used to construct this *white*-specific gRNA were the following (sequences corresponding to the gRNA are underlined):

white_gRNA_top: TGCA<u>ATACCATTCCTGCTCTTTGG</u>
white_gRNA_bottom: AAAC<u>CCAAAGAGCAGGAATGGTAT</u>

Wild-type flies were reared in cages and allowed to lay embryos on the agar plate for 30 min. The agar plate was retrieved, and the embryos were gently moved to the center of the plate using a paintbrush. The agar plate was inserted into the agar plate holder and the automated microinjection was performed. Next, the embryos were left on the agar plate for 3–4 days. The embryos that developed into larvae were counted and transferred into vials with food. Once the larvae developed into flies, the male and female files were transferred into separate vials. Then, the male flies were crossed with female *white* (*w*) flies, and the female flies were crossed with male *w* flies. The resulting progeny from the crosses were then manually scored for transgenesis or a mutant phenotype. Briefly, the flies were put on a $CO_2$ pad to immobilize them and observed under a fluorescence dissection microscope. For piggyBac and PhiC31-mediated transgenesis, scoring relied on the presence or absence of GFP expression in the eyes or thorax, respectively. For CRISPR mutagenesis, eye color phenotypes were scored (Supplementary Fig. 3).

## Molecular characterization of CRISPR white alleles

The molecular characterization of the *white* alleles generated by CRISPR–Cas9 mutagenesis was carried out by PCR analysis of single flies and Sanger sequencing of the amplified DNA fragments. The following primers were used to amplify a 435 bp DNA fragment of the *white* gene containing the *white* gRNA target sequence.

CRISPR_w_1F: GGGCAAAACGATTGCCGAAT
CRISPR_w_2R: GGAGAAGTTAAGCGTCTCCAGG

## Barcoding experiment

### Preparing diverse TaG-EM barcode plasmid library

A gBlock (Integrated DNA Technologies, IDT) containing a 14 bp randomer sequence was cloned into pJFRC12 as previously described (Mendana *et al.* 2023). Briefly, seven independent reactions were set up where the *Eco*RI (NEB), *Psi*I (NEB) digested gBlock was ligated into the *Eco*RI (NEB), *Psi*I (NEB) digested pJFRC12–10XUAS-IVS-myr::GFP backbone using the following reactions conditions: 4 μL T4 ligase buffer (10×) (NEB), 20 μL plasmid backbone DNA (0.005 pmol), 5 μL digested gBlock DNA (0.03 pmol), 2 μL of T4 DNA ligase (NEB), and 9 μL nuclease-free water were mixed and incubated at 22 °C for 2 h. Two microliter of each of the ligation reactions was transformed into 50 μL of TOP10 competent cells (Invitrogen), and the cells were incubated on ice for 30 min, then heat shocked at 42 °C for 30 s, and incubated on ice for 5 min. SOC (250 μL) was added and the cells were plated on Lysogeny broth (LB) + Ampicillin plates and incubated overnight at 37 °C. After overnight incubation, colonies (~10,000–20,000) were resuspended in LB and scraped off the plates and pooled. Plasmid DNA was isolated from the pooled transformant cells using a High-Speed Maxi Prep kit (Qiagen).

In order to confirm successful cloning, PCRs either flanking the gBlock barcode insertion (SV40_5F/SV40_preR primers) or specific to the gBlock barcode insertion (B2_3'F_Nextera/SV40_preR_Nextera primers) were carried out. Primer sequences:

SV40_5F: CTCCCCCTGAACCTGAAACA
SV40_preR: ATTTGTGAAATTTGTGATGCTATTGCTTT
B2_5F_Nextera: TCGTCGGCAGCGTCAGATGTGTATAAGAGAC AGCTTCCAACAACCGGAAG*TGA
SV40_preR_Nextera: GTCTCGTGGGCTCGGAGATGTGTATAAG AGACAGATTTGTGAAATTTGTGATGCTATTGC*TTT

The following reaction conditions were used: 1 μL of template DNA (1:10 dilutions of pJFRC12 control or pJFRC12 + gBlock sample), 1 μL primer 1 (see above), 1 μL primer 2 (see above), 7 μL nuclease-free water, 10 μL KAPA HiFi 2× ReadyMix. The following cycling conditions were used: 95 °C for 5 mins, followed by 30 cycles of 98 °C for 20 s, 60 °C for 15 s, 72 °C for 30 s, followed by 72 °C for 5 min.

### Injecting TaG-EM barcode library and isolating transgenic lines

The robot was used to perform 1,713 injections in the *Drosophila* strain P{y[ + t7.7] = nanos-phiC31\int.NLS}X, y[1] sc[1] v[1] sev[21]; P{y[ + t7.7] = CaryP}attP2. The optimized parameters for depth, speed, and volume were used for the injections. Male flies from injected embryos were crossed to *w*− virgin females and their male progeny was scored for the *w*+ transformation marker. A total of 99 injected males gave rise to transgenic progeny.

### Single fly genomic DNA extraction

In a first experiment up to 10 *w*+ males derived from each unique injected fly were allocated to individual wells of deep-well (500 mL) 96 well plates together with a 3.97 mm (5/32 inch) stainless steel ball bearing (BC Precision, part number 532BCSS30) (Lang *et al.* 2015) and extracted using a magnetic bead-based extraction protocol adapted from a procedure developed by Huang *et al.* (2009). In a follow-up experiment additional progeny from seven injected flies were extracted and sequenced as described below. After allocating into plates, flies were stored at −20 °C prior to DNA extraction. One hundred microliter of Buffer A (100 mM Tris–HCl pH 8.0, 100 mM EDTA, 100 mM NaCl, 0.5% SDS) and 1.5 μL Monarch RNAseA (NEB) was added per well. Plates were sealed using a plate heat sealer and a foil seal and covered with an additional B seal (Thermo). Next, plates were shaken on a TissueLyser II (Qiagen) at 900 rpm (15 rps) for 2 min and then incubated 37 °C for 10 min in a dry incubator. Ten microliter of 20 mg/mL Proteinase K (NEB) was added to each well and samples were incubated at 65 °C for 30 min. Next, 400 μL of Buffer B (1.42 M Kac, 4.28 M LiCl) was added to each well and plates were sealed and mixed well by inverting multiple times, then incubated at 4 °C for at least 10 min and up to ~2 h. The plates were then centrifuged at 3,700 rpm for 5 min at room temperature. Four hundred microliter of the supernatant was transferred to 1 mL deep-well 96 well plates loaded with 500 μL of GNB magnetic beads (GNB magnetic bead recipe: 10 mL Sera-Mag caroboxylate-modified SpeedBeads (Sigma) washed three times in 10 mL of 1xTE, 90 g PEG-8000, 100 mL 5 M NaCl, 5 mL Tris–HCl (pH 8.0), 1 mL 0.5 M EDTA (pH 8.0), sterile water up to 500 mL total volume) using a VIAFLO 96 (Integra) with the pipet and mix setting (mixing 600 μL, five times), then incubated at room temperature for 5 min. Plates were centrifuged at 3,700 rpm for 2 min at room temperature to pellet most of the beads and placed on a magnet until

all the beads pelleted. With the plates still on the magnet, the supernatant was removed and discarded. With the plates still on the magnet, beads were washed twice with 500 μL of 75% ethanol and then incubated for 5 min at room temperature to dry the bead pellet. Fifty microliter of nuclease-free water was added to elute the DNA from the beads and the plates were vortexed for 1 min at 1,400 rpm on a plate mixer. Plates were again placed on the magnet to collect the beads and the supernatant containing the extracted DNA was transferred to a new 96 well plate and stored at −20 °C.

### TaG-EM barcode library preparation and sequencing

To sequence the extracted single fly DNA and the injected barcode library, a two-step amplification and indexing PCR process was used to generate sequencing libraries (Mendana et al. 2023, Apr 1). The following primers were used to amplify the TaG-EM barcodes:

Forward primer pool: four primers with frameshifting bases to increase library sequence diversity in initial sequencing cycles were normalized to 10 μM and pooled evenly to make a B2_3′ F1_Nextera_0-6 primer pool:

B2_3′F1_Nextera: TCGTCGGCAGCGTCAGATGTGTATAAGAGACAGCTTCCAACAACCGGAAG*TGA

B2_3′F1_Nextera_2 TCGTCGGCAGCGTCAGATGTGTATAAGAGACAGAGCTTCCAACAACCGGAAG*TGA

B2_3′F1_Nextera_4 TCGTCGGCAGCGTCAGATGTGTATAAGAGACAGTCGACTTCCAACAACCGGAAG*TGA

B2_3′F1_Nextera_6 TCGTCGGCAGCGTCAGATGTGTATAAGAGACAGGAAGAGCTTCCAACAACCGGAAG*TGA

The following reverse primer was used:

SV40_pre_R_Nextera: GTCTCGTGGGCTCGGAGATGTGTATAAGAGACAGATTTGTGAAATTTGTGATGCTATTGC*TTT

The following amplification reactions were set up:

5 μL template DNA, 1 μL 10 μM B2_3′F1_Nextera_0-6 primer pool (10 μM), 1 μL SV40_pre_R_Nextera (10 μM), 10 μL 2× KAPA HiFi ReadyMix (Roche), 3 μL nuclease-free water. Reactions were amplified using the following cycling conditions: 95 °C for 5 min, followed by 35 cycles for single fly extractions (or 25 cycles for injected barcode library DNA) of 98 °C for 20 s, 60 °C for 15 s, 72 °C for 30 s, followed by 72 °C for 5 min. Next, these PCR reactions were diluted 1:100 in nuclease-free water and amplified in the following indexing reactions: 3 μL PCR 1 (1:100 dilution), 1 μL indexing primer 1 (5 μM), 1 μL indexing primer 2 (5 μM), and 5 μL two times KAPA HiFi ReadyMix (Roche). The following indexing primers were used (X indicates the positions of the 8 bp indices):

Forward indexing primer: AATGATACGGCGACCACCGAGATCTACACXXXXXXXXTCGTCGGCAGCGTC

Reverse indexing primer: CAAGCAGAAGACGGCATACGAGATXXXXXXXXGTCTCGTGGGCTCGG

Reactions were amplified using the following cycling conditions: 95 °C for 5 min, followed by 10 of 98 °C for 20 s, 55 °C for 15 s, 72 °C for 1 min, followed by 72 °C for 5 min. Amplicons were then purified and normalized using a SequalPrep normalization plate (Thermo Fisher Scientific), followed by elution in 20 μL of the elution buffer. An even volume of the normalized libraries was pooled and concentrated using 1.8× AmpureXP beads (Beckman Coulter). Pooled libraries were quantified using a Qubit dsDNA high sensitivity assay (Thermo Fisher Scientific and libraries were normalized to 2 nM for sequencing. The libraries were denatured with NaOH and prepared for sequencing according to the protocols described in the Illumina MiSeq Denature and Dilute Libraries Guides and loaded on a MiSeq at 8 pM with 15% PhiX. Sequencing data for this project is available through the National Center for Biotechnology Information (NCBI) Sequence Read Archive BioProject PRJNA944637.

### TaG-EM barcode data analysis

Demultiplexed fastq files were generated using bcl-convert. TaG-EM barcode data was analyzed using custom Python scripts and BioPython (Cock et al. 2009). For the single fly barcode analysis, leading primer sequences and trailing sequences were trimmed using cutadapt (Martin 2011) and the remaining 14 bp barcode sequences were analyzed to determine the sequence composition at each site. If >80% of the reads at each site corresponded to a single base, the base was called, otherwise the sample was considered to have mixed sequence. Samples with fewer than 500 sequencing reads were not included in the analysis. The number of unique barcodes observed deriving from each injected fly that generated a $w+$ transgenic were then counted. For the barcode library analysis, the leading primer sequences and trailing sequences were trimmed using cutadapt (Martin 2011) and the number of unique 14 bp barcodes observed at different levels of library subsampling was plotted.

## Zebrafish microinjection
### Zebrafish embryo collection and zebrafish care

Zebrafish embryos were collected at the University of Minnesota Zebrafish Core Facility using standard collection procedures (Westerfield 2000). All care and welfare for the animal met NIH animal care standards and were approved by the University of Minnesota Institutional Animal Care and Use Committee (IACUC). Zebrafish parent clutches and their embryos were kept at 28 °C in embryo media (EM) (Westerfield 2000). Once collected, zebrafish embryos were randomly distributed across a 9 cm diameter agar plate, which can be made using any standard zebrafish transplantation mold (McKee and Wingert 2016). Growing fish post-injection was done in the University of Minnesota Zebrafish Core Facility.

### Zebrafish embryo survival analysis

For survival analysis, zebrafish embryos were examined at 1, 3, 24 h, 2, 3, 4, and 5 days post-injection. For the first three time points, the embryo was considered alive if it was developing and showing signs of development within the chorion between consecutive time points. Two- to four-day time points were when the embryos were expected to begin hatching. At the day 5 time point, an embryo was considered alive if it hatched, was able to swim upright in the water column, and had proper cardiac development, tail musculature development, fins, and a swim bladder. Any fish that did not match these criteria was considered abnormal and were not counted among the surviving fish. Experimental data was excluded from analysis if the control group's day 5 survival rate was below 80. Zebrafish embryo survival rates stabilize after day 3. As a result, only survival data up to day 3 is presented here. All robotic microinjection optimization experiments involving zebrafish embryos were performed with at least six replicates on three independent different days with two trials on each day. Each trial contains at least $n = 45$ zebrafish embryos.

## Cryopreservation and laser nanowarming experiment

The cryopreservation and laser nanowarming technique used was similar to our previous study (Khosla *et al.* 2017, 2020, 2021). Zebrafish embryos were microinjected with 10 nL of CPA and GNRs (80% propylene glycol (PG) and 20% methanol (MeOH) with 100 μg/mL GNR) at the center of the yolk of zebrafish embryo at high cell stage. For this study, the standard laser-absorbing PEG-coated GNRs (nanoComposix Inc, San Diego, CA) were used. Both robotic and manual microinjections were performed for each experiment. After microinjection the embryos were cultured inside an incubator at 28 °C for 2–4 h. Then zebrafish embryos were placed into a precooling bath containing 2.7 M PG, 1.2 M MeOH, and 0.5 M Trehalose (Tre) for 5 min to dehydrate perivitelline fluid. After the precooling bath, zebrafish embryos were placed on the cryotop to begin rapid cooling and laser nanowarming. Once the zebrafish embryo was on the cryotop, a droplet (1 μL) containing PG (2 M), MeOH (1.2 M), and GNRs (300 μg/mL) was placed around the embryo. Then the zebrafish embryo along with the droplet was held in liquid nitrogen for at least 1 min for rapid cooling, and to achieve equilibrium at −196 °C. To initiate laser nanowarming, a zebrafish embryo was brought into the laser's focus and the laser pulse was initiated (i980w, LaserStar Inc, Orlando, FL) using previously determined optimal settings required for a laser fluence rate of $1.1 \times 10^9$ W m$^{-2}$, to be 300 V power, 1 ms pulse time. This entire process was recorded using an overhead microscopic camera, and the video was used to evaluate successful vitrification, partial vitrification, or unsuccessful vitrification. After the laser nanowarming process, zebrafish embryos were placed back into post-warming bath for 20 min (half the strength of precooling bath for 10 mins, one-fourth the strength of precooling bath for 10 min) followed by washing in EM to remove any excess CPA.

## Cryopreservation and laser nanowarming experiment survival rate comparison

The survivability metric was adapted from Janik *et al.* (2000). Each experiment was performed at least five times on five independent different days. Whenever the control group's day 5 survival rate was below 80%, the experimental data from that group was excluded from the analysis. To compare the effect of robotic microinjection with the manual microinjection, the survival rate of zebrafish embryos was calculated after each step. For each group, survival rate was calculated for each day through day 5. The first group contains microinjected zebrafish embryos with CPA and GNRs. The second group contains zebrafish embryos with microinjection followed by a precooling and post-warming bath. The third group contains zebrafish embryos with complete cryopreservation and laser nanowarming experimental procedures. Successful, partial, and unsuccessful vitrification rates of zebrafish embryos were also compared between robotic and manual microinjected embryos using videos recorded during the rapid cooling step.

## Results

### Robot construction

Manual microinjection sessions typically involve an initial transfer process wherein embryos laid on a plate or container are first collected, carefully secured on a secondary plate, and in some cases aligned, for visualization under a microscope for microinjection. In time-critical experiments, such as transgenesis in Drosophila where germ cells need to be targeted within 90 min of embryo collection (Rubin and Spradling 1982; Raff and Glover 1989), the

transfer and alignment process, as well as related processes such as dechorionation and dehydration, are rate-limiting steps. Similar time constraints apply to microinjection in zebrafish as well, where optimal transgenesis and cryopreservation are achieved by microinjecting zebrafish embryos during the 1–4 cell stage (Rosen *et al.* 2009) and high cell stage, respectively (Kimmel *et al.* 1995; Khosla *et al.* 2017; Guo *et al.* 2024). Performing microinjection on embryos located on petri dish with minimal sample preparation or manipulation could enable increased scale by eliminating rate-limiting and labor-intensive processes of sample handling prior to injection. Accordingly, we constructed a robot (Fig. 1, a and b) capable of imaging an agar plate containing embryos at multiple scales and perspectives. A commercially available digital single lens reflex (DSLR) camera is used to acquire an image of the whole agar plate at the macroscale to visualize all embryos on the agar plate. A pair of custom-built microscopes are used to stereoscopically image each embryo to visualize the microscale anatomy in 3D. A translation stage manipulates the plate between the DSLR and the dual view microscopes and is further used to guide the microinjection location detected within each embryo to the tip of a stationary microinjection micropipette tip. A central computer interfaces with each of these components and further controls the ejection of the microinjectant from the micropipette.

### Robot operation

The operation of the robot progresses in two stages. In the training stage (Fig. 1c, I), several images of agar plates containing hundreds, or thousands of embryos are acquired under varying imaging conditions to account for differences in image quality and illumination using the DSLR (Fig. 1c, I, i). Embryos in these images are manually annotated to compile a dataset used to train convolutional neural networks (CNNs) (see *Methods* for more details) to detect image features indicative of embryos (Fig. 1c, I, ii–iii). Similarly, dual-view microscopes are used to acquire images of the microinjection micropipettes and individual embryos (Fig. 1c, II, iv). The microscopic images are annotated with the location of the tips of the micropipettes and the desired locations within the embryos where the microinjection needs to be performed (Fig. 1c, I, v–vi). In Drosophila, this is at the posterior end of the embryo, distal to the dorsal appendages, whereas in zebrafish, at the center of the yolk (a central darker portion of the embryo).

Once the ML models are trained, they can be used in subsequent microinjection sessions (Fig. 1c, II). Each agar plate is first imaged using the DSLR and the first ML model is used to detect the x and y locations of each embryo on the plate (Fig. 1c, II, i–ii). This information is used to position each embryo beneath the dual view microscopes to simultaneously image both the micropipette and the embryo from two perspectives (Fig. 1c, II, iii). The second ML model is then used on these stereoscopic images to detect the location of the micropipette tip and the microinjection point within the embryo in 3D space (Fig. 1c, II, iv). The robot then guides the embryos to collocate the microinjection point with the micropipette tip followed by controlled ejection of nanoliter volumes of microinjectants (Fig. 1c, II, v). This process is sequentially executed until all embryos detected on the agar plate are microinjected. Supplementary Videos 1 and 2 shows robotic microinjection of Drosophila and zebrafish embryos.

### Machine learning models for object detection at multiple scales and perspectives

A ML model using a Faster R-CNN algorithm (Ren *et al.* 2017) was used to detect single isolated Drosophila embryos located within

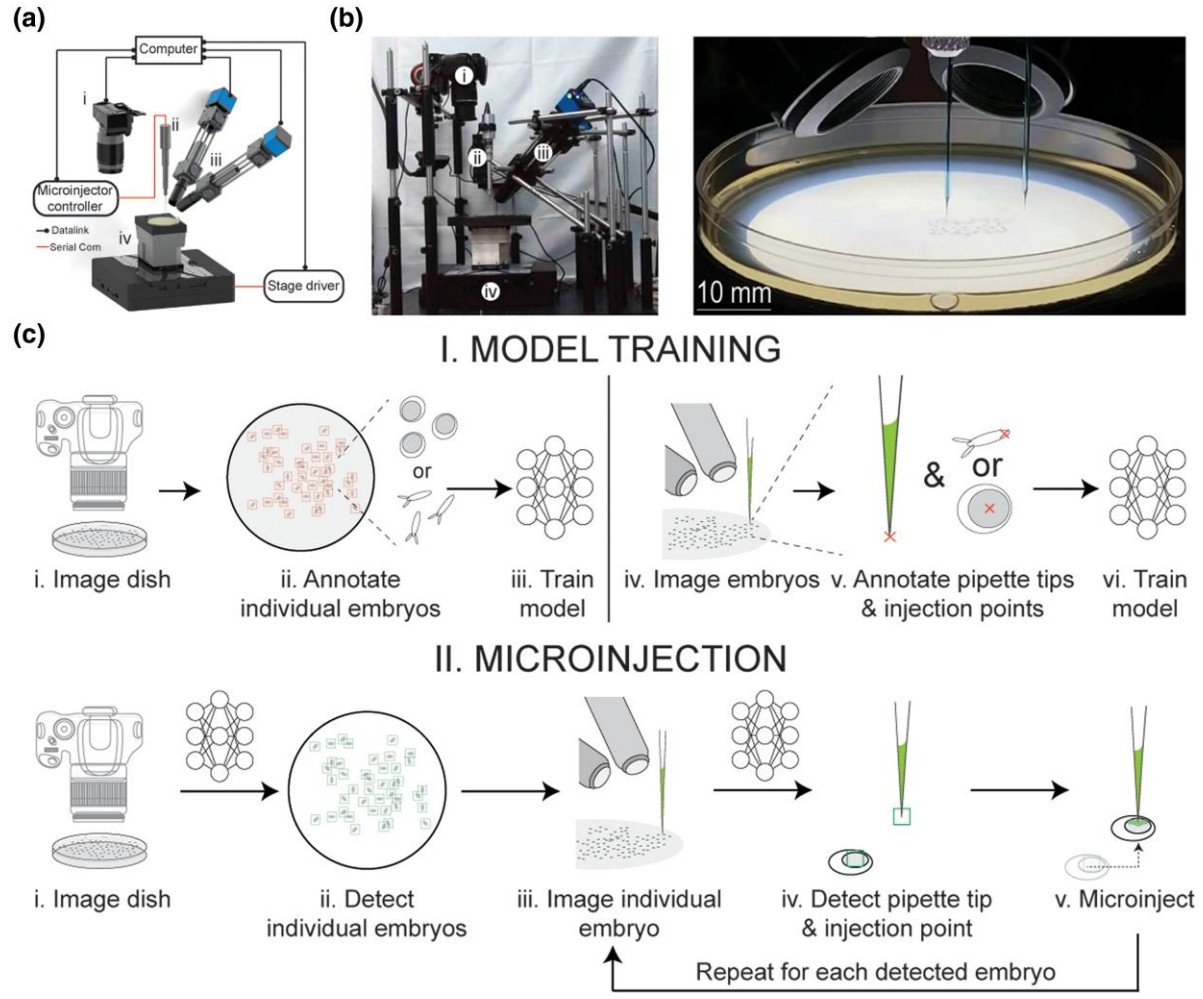

**Fig. 1.** Robot hardware and operation: a, b) (i) DSLR camera, (ii) microinjector controller, (iii) inclined microscopes, and (iv) XYZ stage. c) Automated microinjection procedure for Drosophila and zebrafish. (I, i) Macroscale imaging of the agar plate using the DSLR camera, (I, ii) annotating all individual embryos from macroscale image, (I, iii) training a model using the annotated data, (I, iv) imaging embryos and micropipette tip using the inclined microscopes, (I, v) annotating images acquired from the inclined microscope, and (I, vi) train a model using the annotated data. (II, i) Macroscale imaging of the agar plate using the DSLR camera, (II, ii) using the trained model for the agar plate to detect individual embryos, (II, iii) imaging current embryo and micropipette tip using the inclined microscopes, (II, iv) using the trained model on data acquired by the inclined microscopes, detecting the micropipette tip and microinjection point on the embryo, (II, v) microinjecting the embryo with a solution in the micropipette.

the macroscale image captured by the DSLR, and a second ML model was used to detect the micropipette tip and the microinjection point location at the posterior end of the Drosophila embryos within the images captured using two inclined microscopes (Fig. 2, a–c). We computed the AP, a measure of the accuracy of a trained ML model for each of the ML models used for object detection. All three ML models had an AP above 90% (Fig. 2d). As the ML models are utilized sequentially, cumulatively, the robot can successfully target ~81% of the embryos on a given agar plate for microinjection. As for the other 19%, failures occur when there are false positive detected such as debris and/or bubbles on the agar plate or when there are false negatives such as embryos that are too close to each other and cannot be detected by the ML model. Thus, the ML models were highly accurate in detecting a large fraction of the embryos located on the agar plate in the macroscale DSLR images, providing x and y location estimates of each candidate embryo. When imaged using the two inclined microscopes, each bounding box indicated the microinjection point within a ~$120 \times \sim 159 \ \mu m^2$ area in each perspective image. In comparison, the posterior end of the embryo where the germ cells are located is ~$45 \times \sim 126 \ \mu m^2$. Thus, we can pinpoint a target location

within each embryo for microinjection with accuracy sufficient to subsequently target the embryo for microinjection.

## Machine-vision guided robotic microinjection into Drosophila embryos

Manual microinjection for transgenesis in Drosophila typically involves collecting embryos, removing the outer shell of the embryos (dechorionation), lining each embryo up on a glass slide, and then using a manual microinjection system to inject the embryos at the posterior pole where the germ cells will form. In contrast, we designed an automated microinjection system to directly microinject freshly laid embryos with intact chorions. This is advantageous as it minimizes the handling of the embryos, and the dorsal appendages provide a clear visual landmark for the anterior of the embryo. With the microinjection robot embryos are injected from above, with the micropipette perpendicular to the embryo on the agar plate. In initial proof of concept experiments, we verified that this alternate approach could successfully microinject embryos that subsequently survived the microinjection process (34%, $n = 1{,}318$ embryos).

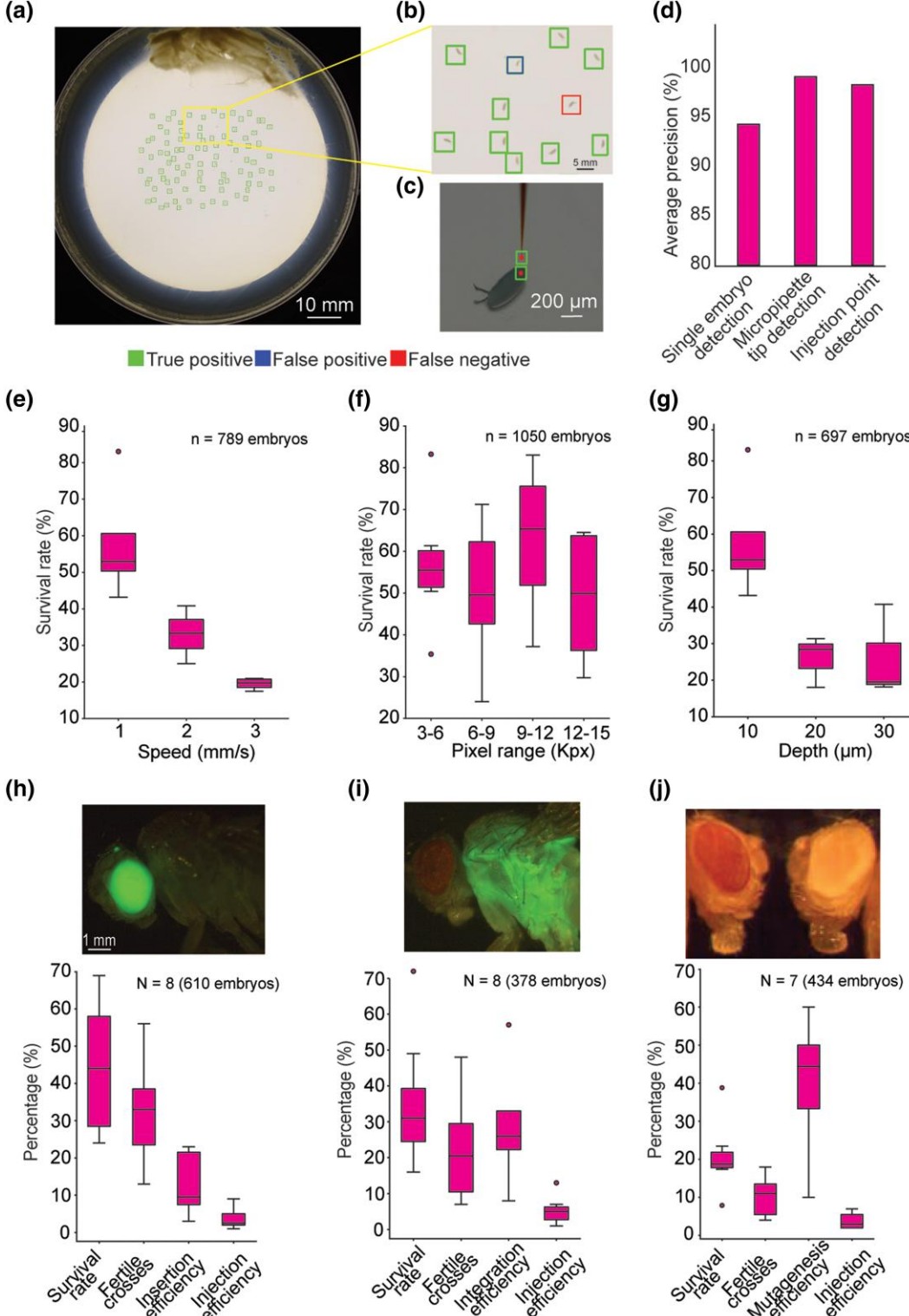

**Fig. 2.** Automated microinjection of Drosophila embryos: a, b) Agar plate containing Drosophila embryos, detected using the Faster R-CNN algorithm. c) Example image showing successful detection of micropipette tip, and microinjection target location within the embryo (posterior pole of a Drosophila embryo). d) AP for Faster R-CNN of single embryo detection in macroscale images as in (a, b), micropipette tip detection as in (c), and microinjection point detection as in (c). e) Drosophila embryo survival rate 3 days after microinjection at three speeds (1–3 mm/s) of micropipette penetration into embryo. f) Survival of Drosophila embryos based on injection volume where the parameters varied from 3 to 15 kPx, where kPx is the number of blue pixels detected for each injection. g) Drosophila embryo survival rate 3 days after microinjection at three depths (10–30 µm) of micropipette penetration into embryo. Total number of embryos injected are shown on the plots. h) piggyBac germline transgenesis with GFP expression in the eyes of the fly. i) PhiC31germline transgenesis with GFP expression in the thorax of the fly. j) CRISPR germline mutagenesis with wild type (wt) fly with red eyes (left) and CRISPR mutant fly with white eyes (right).

Little is known about how the physical act of penetrating the embryo with a micrometer scale micropipette affects the embryo. The execution of the microinjection process using the robot allowed us to systematically evaluate the effect of varying the microinjection depth, speed of the micropipette penetration into the embryo, and volume of solution microinjected into the embryo on the survival rate of Drosophila embryos (Fig. 2, e–g). Microinjections were performed with micropipettes penetrating 10–30 μm into the embryos. As penetration depth was increased, the post-microinjection survival rate decreased with a penetration depth of 10 μm resulting in the optimum depth and highest survival rate (58%, $n = 273$ microinjections, $P = 0.03$, two-sample T-test, Fig. 2g). We next varied the speed of micropipette penetration into the embryo between a range of 1–3 mm/s. We found that the highest survival rate was 58% ($n = 273$ microinjections, $P = 0.07$, two-sample T-test) when the micropipette penetrated the embryo at a speed of 1 mm/s (Fig. 2e). More mechanical damage is likely inflicted on the embryos as microinjection depth and microinjection speed increase, lowering the survival rate. There was no clear trend ($P = 0.60$, two-sample T-test) when the volume of microinjectant was increased within the range of conditions tested, likely because excess microinjectant leaks out of the embryo (Fig. 2f). Thus, using the automated injection system it is possible to rapidly screen microinjection parameters to find optimum conditions for robotically microinjecting Drosophila embryos.

## Automated transgenesis and mutagenesis of Drosophila embryos

We next evaluated whether robotic microinjection could be used to perform successful transgenesis and mutagenesis, techniques which are well established for manual microinjection. Two separate germline transgenesis experiments and one mutagenesis experiment were performed: transposon-mediated transgenesis via piggyBac (Horn et al. 2000), targeted transgenesis using PhiC31 integrase-mediated insertion into an attP landing site (Groth et al. 2004), and CRISPR/Cas9 mutagenesis (Gratz et al. 2015). In the PiggyBac transgenesis experiments, the post-microinjection larval survival rate was 41% ($N = 8$ trials, 610 embryos, Supplementary Fig. 5), with an insertion efficiency (percentage of independent fertile crosses where a transgenic fly carrying the PiggyBac transposon was observed) of 9%, and a microinjection efficiency (percentage of independent transgenic flies compared to the number of microinjected embryos) of 3% (Fig. 2h). Similarly, for PhiC31-mediated integration, the overall survival rate was 35% ($N = 8$ trials, 378 embryos, Supplementary Fig. 5), integration efficiency (percentage of independent fertile crosses where a transgenic fly was observed) was 23%, and overall microinjection efficiency was 5% (Fig. 2i). Lastly, for CRISPR/Cas9 mutagenesis, we injected an sgRNA plasmid targeting the *white* gene into a nos-Cas9 background and obtained an overall survival rate of 22% ($N = 7$ trials, 434 embryos, Supplementary Fig. 5), with a mutagenesis efficiency (percentage of independent fertile crosses where a *white* mutant fly was present) of 40%, and overall microinjection efficiency of 4% (Fig. 2j). The performance of the robot for piggyBac and PhiC31-mediated transgenesis is comparable to efficiencies obtained for manual microinjection by an experienced commercial provider (Supplementary Fig. 6) (Gohl et al. 2011). In all these experiments, there was some variation in survival rates and insertion/integration/mutagenesis efficiencies, we attribute this to many biological factors such as older flies or unfertilized embryos in specific embryo collection.

## MV guided robotic microinjection generalizes to zebrafish

The process of microinjection is fundamentally similar regardless of the size and shape of organism being targeted. We next asked if the robot used for Drosophila microinjections could be used to microinject Zebrafish embryos by training a new set of ML algorithms. Corresponding ML models using YOLO v4 (Bochkovskiy et al. 2020) were used to detect single isolated zebrafish embryos on an image captured using a DSLR camera, micropipette tip location and microinjection point location at the center of the yolk of the zebrafish embryo on an image captured using a microscope (Fig. 3, a–c). All three ML models had an AP above 90% (Fig. 3d). As the ML models are utilized sequentially, cumulatively, the robot can successfully target ~81% of the embryos on a given agar plate for microinjection. Thus, the ML models were highly accurate in detecting a large fraction of the embryos located on the agar plate in the macroscale DSLR images, providing x and y location estimates of each candidate embryo. The ML model using the YOLOv4 algorithm used in zebrafish microinjection experiments defined a bounding box ~375 μm × ~330 μm within each microscope image, which encompassed most of the yolk, the target site for microinjection in this case.

We microinjected EM into zebrafish (Westerfield 2000) during the high cell development stage to evaluate robotic microinjection parameters that optimize post-injection survival. Embryo survival was monitored up to 3 days after microinjection, compared with survival rate of uninjected embryos placed in the same agar plate. All microinjections were performed in the yolk of the embryo. Low speed of penetration of micropipette into the embryo resulted in a higher survival rate but also had lower fraction of the microinjection attempts resulting in successful penetration of the embryo (Fig. 3e). We found that 0.5 mm/s resulted in the highest success rate for penetrating embryos at 75.14% and 48.06% ($n = 251$ microinjections, $P = 0.60$, two-sample T-test) surviving the microinjection attempt. We hypothesize that the low speed of penetration of microinjection needle results in less physical damage to the chorion of the zebrafish embryo, resulting in higher survival rates, but the low speed of penetration generates lower force to penetrate the chorion and yolk of the zebrafish embryo which results in higher fraction of unsuccessful embryo penetrations. As the volume of microinjectant was increased, the survival rate of zebrafish embryos decreased. Microinjecting greater than 10 nL ($n = 307$ microinjections, $P = 0.43$, two-sample T-test) of EM resulted in a substantially lower normalized survival rate of zebrafish embryos (Fig. 3f). In contrast to Drosophila, where excess microinjectant can leak out of the microinjection site, the microinjectant was introduced inside the yolk, a fully confined sac within the zebrafish embryo. Thus, increasing the volume of microinjectant may result in increased pressure within the embryo (Hagedorn et al. 1998; Janik et al. 2000), thereby leading to decreased survival. The rate at which the microinjectant was introduced into the embryo also has a minor effect on the survival of zebrafish embryos (Fig. 3g), with a microinjection rate of 2 nL/s for a volume of 10 nL resulting in the best normalized survival rate ($n = 307$ microinjections, $P = 0.17$, two-sample T-test). We hypothesize that a high rate of microinjection results in higher osmotic shock which leads to low survival rates, while a low rate of microinjection results in a low survival rate of zebrafish embryos, possibly due to the longer duration of insertion of the micropipette inside the embryo.

We next performed two somatic transgenesis experiments in zebrafish. Plasmids with genes encoding the green fluorescent protein (GFP) (Lee et al. 2012) and red fluorescent protein (RFP)

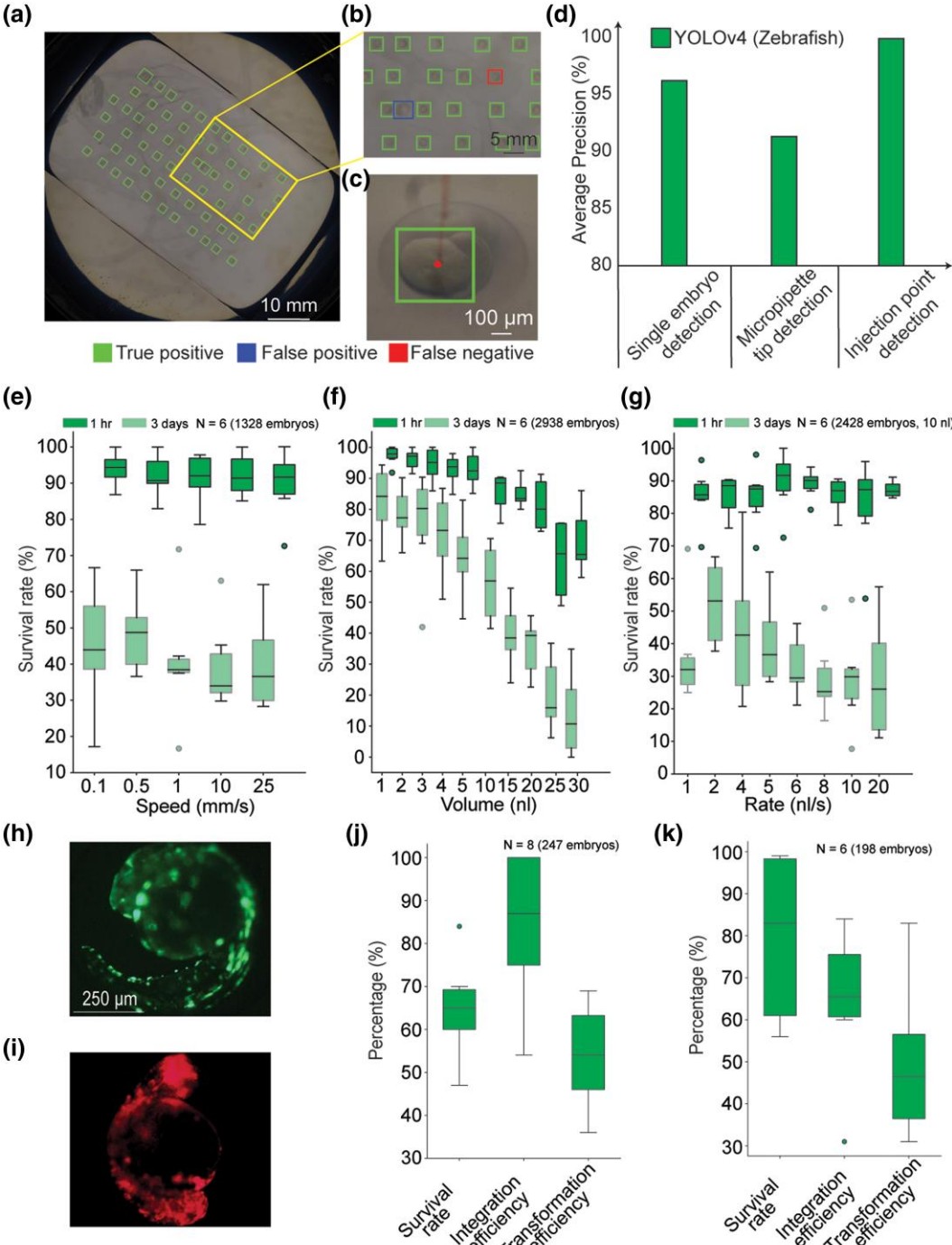

**Fig. 3.** Automated microinjection of zebrafish embryos: a, b) Agar plate containing zebrafish embryos, detected using the YOLOv4 algorithm. c) Example image showing successful detection of the microinjection target location within the embryo (center of the yolk in a zebrafish embryo at the high cell stage). d) AP for YOLOv4 of single embryo detection in macroscale images as in (a, b), micropipette tip detection, and microinjection point detection as in (c). e) Survival of zebrafish embryos at microinjection speeds ranging from 0.1 to 25 mm/s. f) Survival of zebrafish embryos with injection volumes between 1 and 30 nL. g) Survival of zebrafish embryos with microinjections rates between 1 and 20 nL/s. Number of independent trials (total number of embryos injected) are shown on the plots. h, j) GFP somatic transgenesis. i, k) RFP somatic transgenesis. Number of independent trials (total number of embryos injected) are shown on the plots.

(Horstick *et al.* 2015) were injected into 1–4 cell stage zebrafish embryos. GFP or RFP expression was observed 1, 3, and 5 days after microinjection (Fig. 3, h–k). In the same experimental session, somatic transgenesis experiments were carried out via manual microinjection. The automated microinjection procedure consistently showed better transformation efficiency than the manual microinjection procedure (Supplementary Fig. 7). Additionally,

in these experiments, a single micropipette front filled with GFP and RFP plasmid successively, using two storage vials was used. No cross expression of RFP or GFP was found in between the zebrafish embryos. This result indicates that multiple transgenesis experiments can be performed using a single micropipette in a session using the automated microinjection platform (Supplementary Fig. 8).

## Automated microinjection throughput

The results of the transgenesis experiments in both Drosophila and zebrafish embryos indicate that the automated system is capable of robust transgenesis with performance that is comparable to or exceeds manual microinjection. In principle, the robot can perform up to 300 microinjections per hour for Drosophila embryos and up to 600 microinjections per hour for zebrafish embryos. The time required to image the entire plate and detect the embryos is 17 to 20 s. Microinjection of each Drosophila embryo takes approximately 10 s, while the same procedure takes 6 to 12 s for a zebrafish embryo. Variation in speed is a result of varying microinjection parameters required for the experiments, such as volume, rate of microinjectant, and speed of microinjection. The robot construction procedure can be repeated to produce multiple robots, and we found a percentage difference of just 14% between the two independent instruments in terms of injected embryo survival rates (Supplementary Fig. 9).

## Robotic genetic barcoding enables large-scale transgene isolation and measurement of germline insertion rate in Drosophila

To test the efficiency of the automated microinjection robot, we created and injected a pool of >20,000 uniquely barcoded plasmids (Mendana *et al.* 2023) (Supplementary Fig. 10). We injected 1,713 embryos over the course of a few days (Fig. 4a). The survival rate for injected embryos to the larval stage was 50%. Because of the genetic background and transformation marker used in this experiment, we could only score integration events (based on the *w+* marker) in male G0 flies, so 225 surviving injected male flies were crossed to *w–* females yielding 187 fertile crosses. Ninety-nine independent crosses gave rise to transgenic flies

(integration efficiency of 53%, and overall injection efficiency of 5.7%) (Fig. 4b).

Injecting a diverse pool of barcoded DNA constructs allowed us to determine the number of unique transgenes that could be recovered from a multiplexed injection. In addition, the barcodes could be used to determine the number of unique integration events per successfully injected embryo. During embryogenesis, approximately 12–18 pole cells give rise to the future germline formed at the posterior of the embryo (Bownes 1975; Zalokar and Erk 1976; Campos-Ortega and Hartenstein 1985). Each of these germ cells represents a potential target for transgene insertion (and in this experiment, each germ cell was homozygous for the *attP2* landing site). To assess the total number of unique transgenic lines and the number of independent integration events per embryo, we isolated DNA from up to 10 individual *w+* male F1 transgenic flies derived from a male G0 fly that produced transgenic progeny. DNA barcodes were amplified from these extracted DNA samples and sequenced (Fig. 4a). As a control, progeny derived from crossing individual transgenic males were added to two different extraction plates for seven individual lines. While one of the control sequencing reactions failed, 6/6 of the remaining controls produced matching data. In addition, 16 blank samples all produced mixed sequences. Together, these positive and negative controls indicated that the plate-based Illumina sequencing process was successful and specific. Based on the transgene barcode sequences, we identified 386 unique transgenic insertions recovered in this experiment (Fig. 4c; Supplementary Fig. 11).

Surprisingly, single insertions were by far the minority of events and as a rule transgenic flies recovered from the same injection vial (derived from the same injected embryo) represented a diversity of unique insertions, averaging 3.9 independent insertions per successfully injected embryo. Many injected embryos

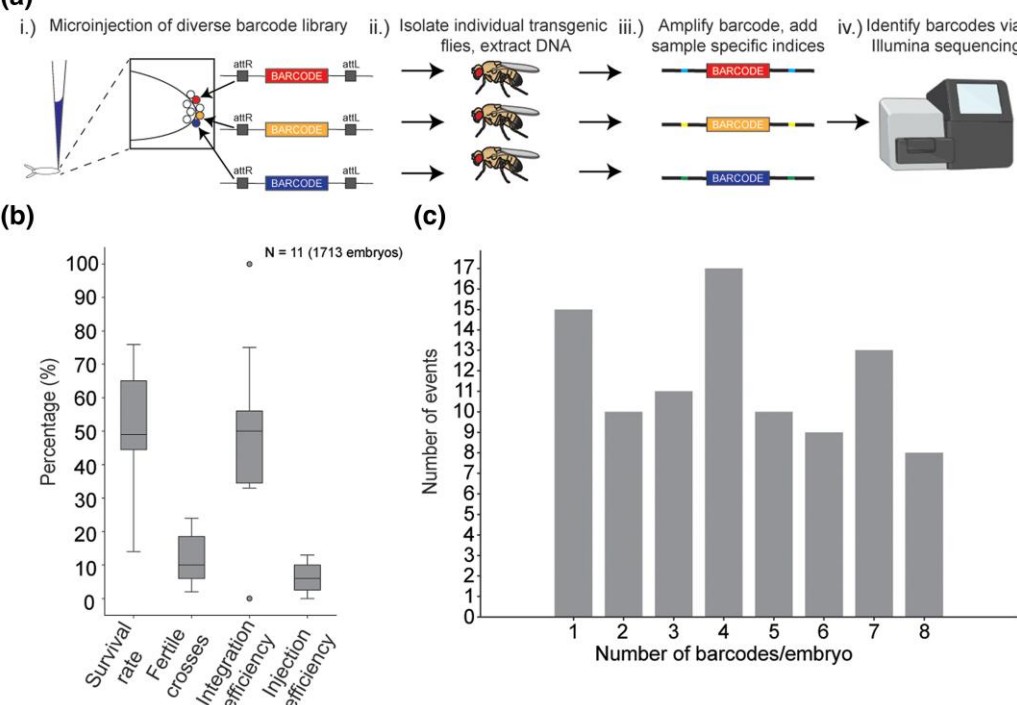

**Fig. 4.** Automated microinjection enables large-scale genetic barcoding experiment: a) (i) microinjection of a plasmid library with >20,000 unique barcodes into Drosophila embryos, (ii) the transgenic flies are then isolated, and their DNA was extracted, (iii) the resulting barcodes are amplified and indexed via PCR, and (iv) the barcodes are sequenced and identified using Illumina sequencing. b) Survival rate, fertile crosses rate, integration efficiency, and injection efficiency for the barcoding experiment. Number of injection plates (total number of embryos injected) are shown on the plot. c) Distribution of the number of distinct barcodes observed for each successfully injected embryo.

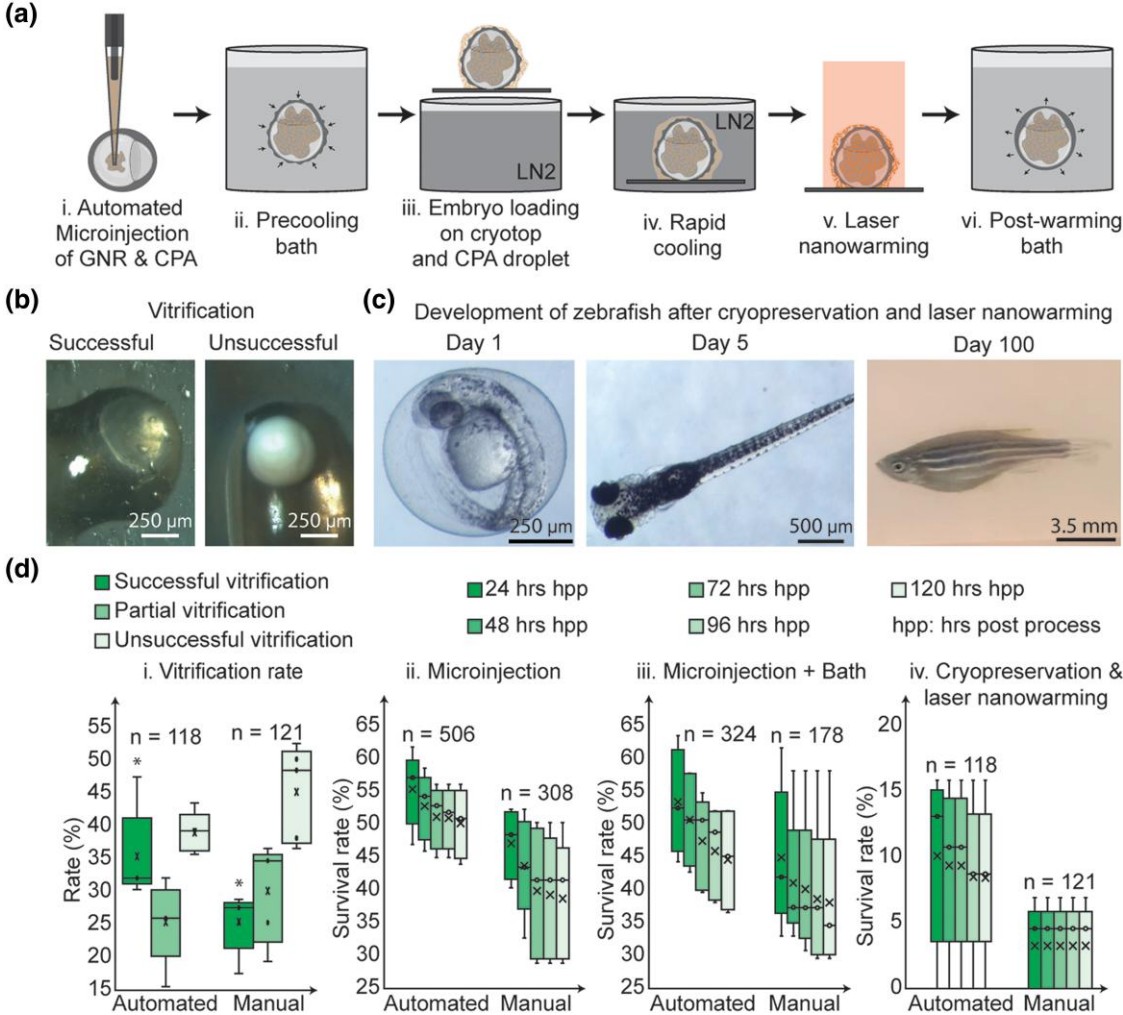

**Fig. 5.** High throughput cryopreservation and laser nanowarming of the zebrafish embryos after automated microinjection of CPA and gold nanorods: a) overview of cryopreservation and laser nanowarming. b) Example of successful and unsuccessful vitrification of zebrafish embryo post-cryogenic stabilization. c) Representative example of zebrafish which survived after cryopreservation and laser nanowarming at days 1, 5, and 100. d) Comparison of survival and vitrification rate between robotic microinjection and manual microinjection after each process. i) Vitrification rate of CPA microinjected embryos. * Indicates $P < 0.05$ (two-sample T-test) (ii) Survivability rates after CPA microinjection. (iii) Survivability rate comparison between robotic ($n = 324$) and manual ($n = 178$) microinjection along with precooling and post-warming baths (Step (a, i, ii, and vi)) (iv) Post-laser nanowarming survival rates of cryopreserved embrbyos. (Step (a, i–vi)).

gave rise to as many as seven or eight unique barcoded lines (Fig. 4c). For a subset of these, we sequenced additional progeny and identified several injected embryos that yielded more than 10 independent barcodes (BZ6_1, BZ8_8, BX_6, BZ5_6, and CB_1; see Supplementary Dataset 1). With this additional sequencing effort, we recovered a total of 410 independent barcoded transgenic lines in this experiment. One successfully injected embryo yielded 15 independent barcodes (CB_1). Thus, in some case we recovered on the order of one independent transformant per pole cell. This demonstrates that with efficient microinjection, many independent integration events can occur within a single embryo and many hundreds of transgenic lines can be generated quickly and efficiently.

## High throughput cryopreservation of whole intact zebrafish embryos

We next sought to leverage the capabilities of the robot to perform microinjection experiments at a scale difficult or impossible for human practitioners to achieve. Cryopreservation plays an important role in preserving tens of thousands of mutants, transgenic, and wild-type zebrafish lines. Cryopreservation at scale could

potentially reduce significant long-term costs and space needs by banking valuable zebrafish lines that are not routinely used. Recently, microinjection of a mixture of cryoprotective agents and gold nanorods have been explored for improving vitrification rates of whole intact zebrafish embryos followed by uniform rewarming of embryos using pulsed infrared lasers (Khosla *et al.* 2017, 2020, 2021). These methods have been successful in reanimating a small subset (3%) (Khosla *et al.* 2020) of embryos which successfully survived to adulthood (Fig. 5a). The issue of a low fraction of surviving embryos could potentially be mitigated by performing microinjection at a larger scale or with higher precision. By using robotic microinjection, we were able to substantially improve the survival of zebrafish embryos post-cryopreservation and laser nanowarming. Embryos automatically microinjected with CPAs were successfully vitrified (Fig. 5b) and subsequently reanimated via laser nanowarming and survived to adulthood (Fig. 5c). Embryos robotically microinjected with CPA have higher rates of survival as compared to manually microinjected embryos (Fig. 5d), significantly improving the rates of vitrification (35.5%, $n = 118$ automated microinjections, 25.13%, $n = 121$, $P < 0.05$, two-sample T-test, Fig. 5, d and i). The post-cryopreservation and laser

nanowarming survival rate after 5 days was 8.5% ($n = 118$ embryos) when the embryos were robotically microinjected, substantially higher than fifth day survival rate of manually microinjected embryos (3.24% $n = 121$ embryos, $P = 0.065$, two-sample $T$-test). Thus, automated microinjection can be used to improve zebrafish cryopreservation procedures through increased scale and precision.

## Discussion

Robotics and automation can remove critical bottlenecks and democratize hitherto skill intensive microbiological procedures such as patch clamping with imaging and computer vision guidance enabling targeted electrophysiology of single cells (Kodandaramaiah *et al.* 2012; 2013; 2016, 2018, 2022; Wu *et al.* 2016; Annecchino *et al.* 2017; Suk *et al.* 2017; Holst *et al.* 2019; Kolb *et al.* 2019; Alegria *et al.* 2020; Joshi *et al.* 2021, 2022; Koos *et al.* 2021). Previous genome-scale microinjection experiments have been performed in a select few labs with significant human and financial resources. In addition to removing constraints related to training and maintaining a workforce with highly specialized skills, automation can potentially enable such large-scale experiments in larger numbers of laboratories. Here, we describe a versatile robotic platform that fully automates the microinjection process for two important model organisms. We use this system to demonstrate several common applications of microinjection, including transposon-mediated and targeted transgenesis and CRISPR/Cas9 mutagenesis.

The automated microinjection system uses multiscale, multi-perspective imaging capabilities, coupled with ML algorithms to detect, within a few micrometers, locations within hundreds of microscale embryos distributed across several cm$^2$. Extending these strategies to real-time detection of an adaptive motion control may in the future enable microinjection into moving subjects, such as *C. elegans* (Ghanta *et al.* 2021) which are difficult to microinject without immobilization. Further, such capabilities can be extended to other microscale interfacing techniques such as patch clamping (Kodandaramaiah *et al.* 2012, 2016), to record activities from single neurons in moving organisms.

In Drosophila, we used the microinjection robot to quickly generate hundreds of unique transgenic lines. From approximately 1,700 injected embryos, using just male injected progeny, we generated 410 unique barcoded transgenes (a number that could likely be doubled if females were also scored). By injecting a diverse pool of barcoded plasmids, we were able to make a novel measurement of the rate of transgene integration in the male germline. Surprisingly, we found that successfully injected embryos gave rise to, on average, at least 3.9 uniquely barcoded transgenic lines, indicating that multiple independent germ cells were targeted by the injected material. Some injected embryos gave rise to more than a dozen independent transgenic lines. These numbers are likely underestimates of the true integration rate as for most lines we only sequenced up to ten progeny per transgene-producing fly. While these rates are undoubtedly specific to our injection conditions (such as amount of injected material) these experiments lay the groundwork for more detailed investigation of germline targeting. In addition, they demonstrate that a surprising and diverse number of transgenic lines can be generated through highly multiplexed microinjection coupled with DNA sequencing. Eliminating the bottleneck of embryonic microinjection provides a path toward routinely conducting large-scale genomic manipulations coupled with next-generation sequencing based readouts in complex animals.

In zebrafish, our data indicate it is not only feasible to microinject CPA into whole embryos for cryopreservation, but it is also possible to substantially improve the overall success rate and vitrification rate. With robotic microinjection, one can perform such operations at orders of magnitude larger scales than manual microinjection (8× speed). While there exists an upper bound for the volume of CPA microinjectant that can be introduced into each embryo (10 nL), alternate strategies, for instance, robotic microdialysis of the yolk followed by microinjection of the equivalent of higher replacement CPA solution can be explored (Jardine and Litvak 2003; Khosla *et al.* 2019; Guo *et al.* 2024). By removing a critical bottleneck in the cryopreservation protocol, the robot opens up the possibility of industrial scale cryobanking of other aquatic species, some of which are critically endangered due to the effects of climate change (Zhan *et al.* 2021).

In summary, we describe a highly versatile automated microinjection system that is capable of targeting embryos with substantially different sizes and constraints. We anticipate that the robot system and approach described here will be readily adaptable to additional organisms and will enable new types of experiments in multicellular animals in conjunction with large-scale sequencing and gene editing tools.

## Data availability

All code for controlling the robot was written in Python and is available at: www.github.com/bsbrl/microinjection. The authors affirm that all data necessary for confirming the conclusions of the article are present within the article, figures, and tables.

Supplemental material available at GENETICS online.

## Acknowledgments

We thank Marc Tye of the University of Minnesota (UMN) Zebrafish Core for their support for the zebrafish studies. We thank our colleagues in the University of Minnesota Genomics Center (RRID:SCR_012413) in particular, Aaron Becker and Dylan Cole for help with DNA sequencing, and Ray Watson for assistance with high throughput fly DNA extraction. Stocks obtained from the Bloomington Drosophila Stock Center (NIH P40OD018537) were used in this study.

## Funding

Funding from the National Institute of Health (NIH) (1R21OD028214, 1R24OD028444), Minnesota Sea Grant, University of Minnesota and National Science Foundation (NSF) EEC 1941543 is gratefully acknowledged. ADA acknowledges the support of a Diversity of Views and Experiences (DOVE) fellowship at the UMN. ASJ acknowledges the support of Minnesota's Discovery, Research, and Innovation Economy (MnDRIVE) fellowship from the University of Minnesota Informatics Institute (UMII).

## Author contributions

ADA and ASJ contributed equally to this work. Order of first authorship was determined by the alphabetical order of last name initials. ADA optimized and implemented the robot for microinjection in Drosophila. ASJ optimized and implemented the robot for microinjection in zebrafish. Full breakdown of author contributions is as follows:

Robot conceptualization: DMG, SBK, ADA, ASJ, BA. Construction and testing: DMG, SBK, ADA, ASJ, BA. Robot

optimization and implementation for Drosophila: DMG, SBK, ADA, JBM, MD, BA. Robot optimization and implementation for zebrafish: DMG, SBK, ASJ, KTS, KK, JB. Robotic barcoding experiments in Drosophila: DMG, SBK, ADA, JBM, MD. Zebrafish cryopreservation experiments: DMG, SBK, ASJ, KTS, KK, JB. Data analysis: ADA, ASJ, KK, JBM, DMG. Writing—original draft: ADA, ASJ, SBK, DMG. Writing—review and editing: All authors.

## Conflicts of interest

ADA, ASJ, DMG, and SBK are co-founders of Objective Biotechnology Inc. which is commercializing the technology described in this manuscript.

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
