## [Peer Review File · Genetics]

High-throughput genetic manipulation of multi-cellular organisms using a machine-vision guided embryonic microinjection robot

Andrew Alegria, Amey Joshi, Jorge Mendana, Kanav Khosla, Kieran Smith, Benjamin Auch, Margaret Donovan, John Bischof, Daryl Gohl, and Suhasa Kodandaramaiah

NOTE: The reviews and decision letters are unedited and appear as submitted by the reviewers.

In extremely rare instances and as determined by a Senior Editor or the EIC, portions of a review may be redacted. If a review is signed, the reviewer has agreed to no longer remain anonymous.

The review history appears in chronological order.

Review Timeline:

Submission Date:	2023-10-09
Editorial Decision:	2023-11-12
Revision Received:	2024-01-02
Accepted:	2024-01-08

November 12, 2023

RE: GENETICS-2023-306540

Dear Dr. Kodandaramaiah:

I am pleased to accept your manuscript entitled "**High-throughput genetic manipulation of multi-cellular organisms using a machine-vision guided embryonic microinjection robot**" for publication in GENETICS, pending minor revision.

Please submit your revision along with a brief description of how you modified the manuscript in response to the reviewers' concerns and suggestions (which can be viewed at the bottom of this email. Please modify the text as recommended by the reviewers, clarifying and adding information as requested. Any questions about the revisions, please do not hesitate to get in touch with me. I expect you should be able to submit a revised manuscript within 30 days. A suitably revised manuscript will be acceptable for publication; I don't expect to send it out for review.

Please ensure that you have included a Data Availability Statement at the end of the Materials and Methods section. Details available at <https://academic.oup.com/genetics/content/prep-manuscript>. The DAS should include the accession numbers or DOIs of any data you have placed in public repositories, describe supplemental material, include applicable IRB numbers, and may include specifications for how to properly acknowledge or cite the data.

When revising the ms., please make an effort to shorten when possible with an eye towards brevity and clarity. We urge authors to heed the advice of Strunk and White: "omit needless words"¹. Follow this link to submit the revised manuscript: Link Not Available

Thank you for submitting this story to Genetics.

Sincerely,

Jill Wildonger
Associate Editor
GENETICS

Approved by:
Kate O'Connor-Giles
Senior Editor
GENETICS

Reviewer comments:

Reviewer #1 (Comments for the Authors (Required)):

Review of Alegria et al "High-throughput genetic manipulation ..."

Alegria et al describes a custom built system for injecting *Drosophila* and Zebrafish embryos. Unlike past efforts to develop automated injection systems for *Drosophila* that have relied on a mechanical solution (e.g. by sucking embryos through something like a flow sorter with a pause to inject), instead the authors cleverly use a binocular set of cameras and machine learning to inject embryos on the same petri plate that females laid eggs on. Unlike typical "technology" papers, the authors do extensive testing of their system, in the process injecting thousands of embryos. The evidence the system works is pretty convincing. The preliminary experiments don't just inject saline into embryos, but actually make several genetic constructs and carefully quantify parameter tweaks that result in a high transformation rate. There is even an interesting novel biological observation showing that given an injection of a pool of constructs a single F0 male can have F1 offspring with several different genetic constructs.

The authors also port the system to Zebrafish. Again, unlike past highly specialized attempts to do automatic microinjection in flies, the Zebrafish work demonstrates their approach is likely general and could be adapted to many different systems. This speaks to why the work is likely of broad interest to the model system genetics community.

I believe the paper is suitable for Genetics. Genetics is a primary journal for the fly, and more generally genetic model, community. A widely available technology like this would benefit that community a great deal.

I have several comments I would like the authors to consider.

Major comments:

The paper should describe the system in sufficient detail that someone could "reproduce" the work of this paper given enough effort. I appreciate that the authors wish to commercialize this technology. And also feel commercialization is the best strategy for actually getting the technology to different labs. The truth is that very few fly labs would have the gumption to try to re-create this system from scratch. I also suspect any commercial platform would likely re-source all the parts from the prototype regardless. So a better description of the parts list doesn't harm that effort. Perhaps this could be accomplished with a supplementary table that is a parts list?

Typically people injecting fly embryos pull their own glass needles. That does not seem to be the case here, and indeed pulling and breaking needles might create variation that would be difficult to accommodate with this machine learning platform. What exactly are the needles being used then? (this speaks to the above point of not really fully describing the system). And how did the authors deal with common problems like clogging? The methods are a little thin for a methods paper.

In the Methods the authors discuss the computing resources for machine learning training. Biologists are often not well-versed in machine learning so a little more detail may be in order. Presumably once the robot is trained to inject fly embryos, that "set of weights" is totally portable (even across robots) and a user need not repeat that process. In this sense the "training" section of the methods is not carefully laid out and could be separated into training the model versus running the model. This separation of the training vs. the running of the model should also be clear in the results. Often-times the computing requirements for training a model are more onerous than those for running a model, this could be more carefully documented. Given a trained model, what processing power is needed to run the machine (clearly this is not trivial as two video streams are being captured and analyzed in real-time)? This would be partially addressed given a fuller description of the complete system (see below).

I looked at the github for the code. The authors should be applauded for the code release. Although the README could be somewhat more complete and discuss installing and running the code. I appreciate that running the code requires the robot. But sometimes a better description of what commands are required to train the machine learning algorithm and what commands are required to run it, would help scientists who wish to adapt a system like this to an unrelated problem. In terms of reproducibility, (at least in theory) given your robot / computer and the git, a practitioner should be able to reproduce the training and an injection experiment. (or at least feel like they could in a conceptual way). For the training required to recognize the embryos which scripts are used and how are they called, what are the input files, what is the output. For training to inject (given an embryo is recognized and its coordinates are known ... this seems like the output of the prior step) it would be useful to address the same questions (input, scripts, output). Given the training is complete, how is that information saved, and used by programs to actually run an injection experiment? What scripts are used to do the actual injection experiment? A more complete README would go a long way.

Minor comment:

Reading through the references I saw several examples of incomplete or poorly formatted citations. It would be worthwhile to go through these by hand and fix them.

One final comment:

The videos of the machine injecting embryos is mind-blowingly cool. The journal should highlight this paper and figure out a way to make one of the videos part of the highlight (which is a trend these days in online publishing).

Reviewer #2 (Comments for the Authors (Required)):

In this manuscript titled "High-throughput genetic manipulation of multi-cellular organisms using a machine-vision guided embryonic microinjection robot" the authors Alegria et al., have developed a novel system that has significantly increased the throughput for microinjections in the *Drosophila* and zebrafish embryos. The authors began by providing details of the setup and its operating steps. The setup includes technological advances in both hardware and software. The authors have used multiple cameras to image the embryos and the injection needle. They then trained a neural network to detect the location of the embryos and microinjection needle. Next, they show the effectiveness and various parameters that affect the survival rate of *drosophila* and zebrafish embryos upon injection. Finally, the authors have used this tool for large-scale genetic barcoding experiments.

This work is well presented by the authors. I have the following major and minor comments for further improvement of this manuscript.

Major comments:

- 1) The authors have used the terms integration efficiency, microinjection efficiency, but it is not very clearly presented. I request the authors to clearly explain these two terms and the difference between them in the main text.
- 2) The authors have not done any statistical tests on some of their datasets. For e.g., the authors claimed that they found 0.5 mm/s as an optimum speed for the microinjection of zebrafish embryos. However, they have not shown that survival rate at 0.5 mm/s is statistically significant over 0.1 mm/s.
- 3) The authors have shown the incredible efficiency of their system. However, they have not discussed why on some occasions the system does not perform correctly (e.g., false positive, false negative trials). What affects the average precision as shown in Fig. 2d? Maybe the authors can add more information about this in the main text.
- 4) I request the authors to add more details to Supplementary Fig. 1 (both in the main text and figure caption).
- 5) Supplementary Fig. 10 and 11 are not mentioned in the main text.

Minor comments:

- 1) There is a typo/grammatical error in the sentence "Microinjection is also can be used to prepare model ..." in the introduction section.
- 2) In the "Robot operation" section, the authors say "Once the ML models are trained, they can be used in subsequent microinjection sessions ...". Can the authors comment on how frequently they need to calibrate the system? Is it days or weeks?
- 3) In the data presented in Supplementary Figure S2, we can see a huge variation in the survival rate and integration efficiency. For e.g., in PiggyBac transgenesis, we see survival rates varying from 70% to as low as 20%. In the same way, the integration efficiency varies from 20% to 2%. I request the authors to comment on why they see such a huge variation in their experimental results.
- 4) In Supplementary Figure S2a, for trial 4, the survival rate bar is zero. But it still has 5% integration efficiency. Did the authors accidentally miss the survival rate bar for trial 4? Or else, how the integration efficiency was calculated in this case.
- 5) There is a typo in the "MV guided robotic microinjection generalizes to zebrafish" section. The authors have swapped Fig. 3e and Fig. 3f in the main text.
- 6) In the "Drosophila image training" section, it is not clear how the authors reached the conclusion of using 0.00003 as a learning rate from the data presented in Supplementary Fig. 8. I request the authors to add more details in the text.
- 7) Please expand Kpx which is used as x-label in Fig. 2f. It is not clear if kPx is a great parameter because the microinjection is in 3D. So, some injected volume that is below the focal plane might not get detected.
- 8) Maybe I missed this, but why are there two microinjection needles in the right panel of Fig. 2b?
- 9) I think the authors meant survival rate (%) as the y-label for the leftmost panel in Fig. 5d.
- 10) The authors have used t-test as a statistical test for the data presented in Supplementary Fig. 6. Since the data shows survival rate, i.e., the embryo either survives or not. Hence there are only two possible outcomes. As a result, it is a binomial distribution. So, it is not clear why the authors chose the t-test.

Reviewer #3 (Comments for the Authors (Required)):

The manuscript by Alegria et al. describes development of an automated microinjection system for injection delivery to *Drosophila* and zebrafish embryos. The manuscript is well written and important to the field, as microinjection remains one of the main bottlenecks for genetic manipulation of model organisms. The experiments described include nice comparisons to manual microinjection of both *Drosophila* and zebrafish embryos. Also described is the use of the injection system for cryopreservation of zebrafish embryos. Overall, the experiments described represent an important step forward for automation of genetic manipulation.

Prior to publication, I would ask the authors to address the following minor points:

81% of the *Drosophila* embryos could only be injected. Why is this so low and are there ways to increase this.

For the *Drosophila* experiments the bounding box is much larger than the desired target in the posterior of the embryo, yet injection is considered to be precise. Please clarify this.

The zebrafish DNA injections should be clarified, and percentages presented in the results section. It is unclear what is meant by the integration efficiency and the transformation efficiency presented in figure 3.

Ideally, the one cell of the one cell zebrafish embryo would be injected. Discussion of this would help the manuscript.

The cryopreservation protocol is extensive. Other than the automation of injection, are there new aspects to the protocol? If so,

this could be better elaborated.

P values are missing for most experiments.

We thank the reviewers for their overall positive assessment and for the constructive comments that have helped us to improve the manuscript. Each of the individual comments from the reviewers is addressed in detail below. Edits to the manuscript have been highlighted as blue text.

Reviewer 1:

1. The paper should describe the system in sufficient detail that someone could "reproduce" the work of this paper given enough effort. I appreciate that the authors wish to commercialize this technology. And also feel commercialization is the best strategy for actually getting the technology to different labs. The truth is that very few fly labs would have the gumption to try to re-create this system from scratch. I also suspect any commercial platform would likely re-source all the parts from the prototype regardless. So a better description of the parts list doesn't harm that effort. Perhaps this could be accomplished with a supplementary table that is a parts list?

We have included a supplemental table that contains a parts list with information about the vendor and catalog number of all the parts used to construct the robot. As noted below, we have made all the code necessary to train models and run the robot available via GitHub.

2. Typically people injecting fly embryos pull their own glass needles. That does not seem to be the case here, and indeed pulling and breaking needles might create variation that would be difficult to accommodate with this machine learning platform. What exactly are the needles being used then? (this speaks to the above point of not really fully describing the system). And how did the authors deal with common problems like clogging? The methods are a little thin for a methods paper.

The micropipettes being used are pulled with a P-1000 Micropipette Puller and are then beveled using a BV-10 Micropipette Beveler. The robot also uses a multi-pipette holder where 4 micropipettes can be installed. Once a pipette is clogged, the operator can switch over to a new micropipette. The following sentence was added to the materials and methods to make this information clear:

Newly added text (lines #451-454): "For the manufacturing of the micropipettes, we used glass aluminosilicate capillaries with filament and pulled the capillaries using a Sutter P-1000 micropipette puller. The micropipettes were then beveled using a Sutter BV-10 micropipette beveler to an ~3 um micropipette tip opening."

3. In the Methods the authors discuss the computing resources for machine learning training. Biologists are often not well-versed in machine learning so a little more detail may be in order. Presumably once the robot is trained to inject fly embryos, that "set of weights" is totally portable (even across robots) and a user need not repeat that process. In this sense the "training" section of the methods is not carefully laid out and could be separated into training the model versus running the model. This separation of the training vs. the running of the model should also be clear in the results. Often-times the computing requirements for training a model are more onerous than those for running a model, this could be more carefully documented. Given a trained model, what processing power is needed to run the machine (clearly this is not trivial as two video streams are being captured

and analyzed in real-time)? This would be partially addressed given a fuller description of the complete system (see below).

The reviewer is correct in that for a specific organism, the model only needs to be trained once and then can then be used on other robots. For this robot, we used a GPU to train our model to detect embryos in the image captured by the DSLR camera for ~15 hours. The model to detect injection point from the images captured by the microscope took ~4 hours to train. While a regular CPU could be used, training time will be significantly increased. See supplementary Figure 8 for plots showing model performance as a function of training time.

4. I looked at the github for the code. The authors should be applauded for the code release. Although the README could be somewhat more complete and discuss installing and running the code. I appreciate that running the code requires the robot. But sometimes a better description of what commands are required to train the machine learning algorithm and what commands are required to run it, would help scientists who wish to adapt a system like this to an unrelated problem. In terms of reproducibility, (at least in theory) given your robot / computer and the git, a practitioner should be able to reproduce the training and an injection experiment. (or at least feel like they could in a conceptual way). For the training required to recognize the embryos which scripts are used and how are they called, what are the input files, what is the output. For training to inject (given an embryo is recognized and its coordinates are known ... this seems like the output of the prior step) it would be useful to address the same questions (input, scripts, output). Given the training is complete, how is that information saved, and used by programs to actually run an injection experiment? What scripts are used to do the actual injection experiment? A more complete README would go a long way.

We have added a more detailed README to the github page which describes hardware and software requirements as well as installation and operating instructions.

<https://github.com/bsbrl/Automated-Drosophila-embryo-microinjection>

5. Reading through the references I saw several examples of incomplete or poorly formatted citations. It would be worthwhile to go through these by hand and fix them.

We thank the reviewer for pointing out these errors. We have fixed these issues.

Reviewer 2:

1. The authors have used the terms integration efficiency, microinjection efficiency, but it is not very clearly presented. I request the authors to clearly explain these two terms and the difference between them in the main text.

We apologize that this was unclear and have added additional information on the calculations used for these metrics in the methods to make this information more accessible.

2. The authors have not done any statistical tests on some of their datasets. For e.g., the authors claimed that they found 0.5 mm/s as an optimum speed for the

microinjection of zebrafish embryos. However, they have not shown that survival rate at 0.5 mm/s is statistically significant over 0.1 mm/s.

We thank the reviewer for this critique and have run statistical tests for the data presented in Figures 2 and 3. Based on our tests, we found that for most of our datasets (including the one the reviewer mentioned) there is not a statistically significant optimum value except for the depth value of 10 μm for *Drosophila* embryo microinjections. The p-values have been added to the text of the paper.

The text describing the specific example mentioned by the reviewer was modified to include the information that this effect was not statistically significant (lines # 271-273): “We found that 0.5 mm/s resulted in the highest success rate for penetrating embryos at 75.14% and 48.06% (n = 251 microinjections, p = 0.60) surviving the microinjection attempt.”

3. The authors have shown the incredible efficiency of their system. However, they have not discussed why on some occasions the system does not perform correctly (e.g., false positive, false negative trials). What affects the average precision as shown in Fig. 2d? Maybe the authors can add more information about this in the main text.

False positives can occur when bubbles or debris on the dish are mistakenly detected as embryos. False negatives can occur when embryos are placed too closely to each other and appear as a non-isolated embryo. The following text has been added to the paper on lines # 184-186 to provide this information:

Newly added text: “The 19% failures occur when there are false positive detected such as debris and/or bubbles on the agar plate or when there are false negatives such as embryos that are too close to each other and can't be detected by the ML model.”

4. I request the authors to add more details to Supplementary Fig. 1 (both in the main text and figure caption).

The caption for Supplementary Figure 1 was updated as follows:

“Supplementary Figure 1: Procedure for microinjection location estimation: In the automated microinjection system, the micropipette remains stationary, serving as a reference point for the control system. The control system output, denoted as dx, dy, dz, is then applied to the XYZ stage of the autoinjector, determining the XYZ location of an embryo. The microinjection point is of primary interest in this context. To detect the microinjection point on an embryo, two inclined microscopes function as sensors in the control loop. These microscopes capture images, which undergo processing through object detection algorithms to identify X and Y pixel locations from each microscope. Utilizing these X and Y pixel locations, the dx, dy, dz in conjunction with the global XYZ embryo locations, are then used to determine the final XYZ locations of the embryos. The control system, using the estimated states and reference locations, generates the next dx, dy, dz output for the XYZ stage. This comprehensive procedure for microinjection point estimation was used in the microinjection of both *Drosophila* and zebrafish embryos.”

We have also added a figure call to line 148 of the main text.

5. Supplementary Fig. 10 and 11 are not mentioned in the main text.

We thank the reviewer for catching this. This was an oversight. The figure calls for these two supplemental figures have been inserted in the appropriate spots.

6. There is a typo/grammatical error in the sentence "Microinjection is also can be used to prepare model ..." in the introduction section.

This typo has been corrected. The sentence now reads:
"Microinjection can also be used to prepare model ..."

7. In the "Robot operation" section, the authors say "Once the ML models are trained, they can be used in subsequent microinjection sessions ...". Can the authors comment on how frequently they need to calibrate the system? Is it days or weeks?

In our current work, the robot has been calibrated every 3 months. Calibration is required when embryos detected from the macroscale image are unable to be detected/not in the field of view of the two inclined microscopes.

8. In the data presented in Supplementary Figure S2, we can see a huge variation in the survival rate and integration efficiency. For e.g., in PiggyBac transgenesis, we see survival rates varying from 70% to as low as 20%. In the same way, the integration efficiency varies from 20% to 2%. I request the authors to comment on why they see such a huge variation in their experimental results.

Variability in survival and integration rate could be due to many biological factors such as older flies or unfertilized embryos in specific embryo collections.

Newly added text (lines # 245-247): In all these experiments, there was some variation in survival rates and insertion/integration/mutagenesis efficiencies, we attribute this to many biological factors such as older flies or unfertilized embryos in specific embryo collection.

9. In Supplementary Figure S2a, for trial 4, the survival rate bar is zero. But it still has 5% integration efficiency. Did the authors accidentally miss the survival rate bar for trial 4? Or else, how the integration efficiency was calculated in this case.

Thanks for pointing this issue. The supplementary figure has been corrected.

10. There is a typo in the "MV guided robotic microinjection generalizes to zebrafish" section. The authors have swapped Fig. 3e and Fig. 3f in the main text.

This has been corrected.

11. In the "Drosophila image training" section, it is not clear how the authors reached the conclusion of using 0.00003 as a learning rate from the data presented in Supplementary Fig. 8. I request the authors to add more details in the text.

We used the default learning rate set in the configuration file for the Faster R-CNN algorithm. We found that this learning rate was sufficient for embryo detection but did not explore the effect of changing this parameter on training performance.

12. Please expand Kpx which is used as x-label in Fig. 2f. It is not clear if kPx is a great parameter because the microinjection is in 3D. So, some injected volume that is below the focal plane might not get detected.

The reviewer makes a valid point. Kpx is the number of blue pixels detected from a given image when a solution with blue dye is injected into the embryo. This is observed from two distinct perspectives and while it is an indirect measure, it provides a way to measure and compare injection volumes.

13. Maybe I missed this, but why are there two microinjection needles in the right panel of Fig. 2b?

One micropipette (in the center) is being used for microinjections. The other micropipette is a “spare” (contained within the multi-pipette holder) in case the micropipette being used gets clogged.

14. I think the authors meant survival rate (%) as the y-label for the leftmost panel in Fig. 5d.

The leftmost panel of Figure 5d shows vitrification rates throughout the cryopreservation process, comparing automated and manual microinjection procedures. Within each section corresponding to automated and manual microinjection, three distinct graphs illustrate the percentage of successful, partial, and unsuccessful vitrification rates observed in the experiments. We believe that expressing these outcomes as Rates (%) rather than survival rates (%) is a clearer and more precise explanation of the data in this panel.

15. The authors have used t-test as a statistical test for the data presented in Supplementary Fig. 6. Since the data shows survival rate, i.e., the embryo either survives or not. Hence there are only two possible outcomes. As a result, it is a binomial distribution. So, it is not clear why the authors chose the t-test.

We thank the reviewer for catching this oversight and have removed the p-value from this figure since the purpose of the figure is to show that survival does not vary much between Robot 1 and Robot 2. Instead, since we have large sample sizes for both robots, we did a percentage difference calculation and obtained a percentage difference of just 14%.

Newly added text (lines # 464-466): The robot construction procedure can be repeated to produce multiple robots, and we found a percentage difference of just 14% between the two independent instruments in terms of injected embryo survival rates (**Supplementary Fig. 6**).

Reviewer 3:

1. 81% of the Drosophila embryos could only be injected. Why is this so low and are there ways to increase this.

Embryos are not injected because either they are not detected by the ML algorithm in the initial macroscale image, the micropipette can't penetrate through the chorion of the embryos, or the injection point is not detected by the ML model. While this number reflects the current state of performance for the ML model and instrument, this could potentially be improved in the future by increasing the number of images in the training dataset and beveling sharper micropipettes. See also response to Reviewer 2 comment 3.

2. For the *Drosophila* experiments the bounding box is much larger than the desired target in the posterior of the embryo, yet injection is considered to be precise. Please clarify this.

The bounding box represents the target area selected while training the ML model to recognize the desired injection site. The actual injection site targeted in this example is at the center of the bounding box. If desired the bounding box used for training could be made smaller or larger depending on the desired injection site and the injection site target location relative to the bounding box could also potentially be adjusted. Finally, for *Drosophila* injections, the injected material diffuses into the area surrounding the injection site, so despite the fact that we can hit the target with high precision, there is probably substantial wiggle room in terms of being able to deliver the payload for a successful injection.

3. The zebrafish DNA injections should be clarified, and percentages presented in the results section. It is unclear what is meant by the integration efficiency and the transformation efficiency presented in figure 3.

See above response to comment 1 by Reviewer 2. Integration efficiency is described on page 6 and we have also added additional clarifying information in the methods section.

4. Ideally, the one cell of the one cell zebrafish embryo would be injected. Discussion of this would help the manuscript.

In the context of transgenesis experiments involving zebrafish embryos, microinjections were carried out either at the one-cell stage, at the intersection of the cell and yolk, or at the center of the yolk. Conversely, in cryopreservation experiments with zebrafish embryos, microinjection specifically targeted the center of the yolk. In this study, to showcase the adaptability of the automated microinjection system, machine learning models were developed to precisely target the center of the yolk in zebrafish embryos. In future, there is potential to improve these machine learning models to target other regions of the zebrafish embryos, thereby expanding the systems' versatility.

5. The cryopreservation protocol is extensive. Other than the automation of injection, are there new aspects to the protocol? If so, this could be better elaborated.

No, only the first step of the protocol, where we inject the embryos automatically is different from the methodology used in Khosla et al 2017. We would also like to note that the automated microinjection of cryoprotectants has now been used in a second publication – Z Guo et al Adv Sci 2023.

6. P values are missing for most experiments.

We have added P-values to the experiments presented in Figure 2 and 3 in the paper text.

January 8, 2024

RE: GENETICS-2023-306540R1

Prof. Suhasa B Kodandaramaiah
University of Minnesota Twin Cities
Mechanical Engineering
111 Church St SE
Minneapolis, Minnesota 55455

Dear Dr. Kodandaramaiah:

Congratulations! We are delighted to inform you that your manuscript entitled "**High-throughput genetic manipulation of multi-cellular organisms using a machine-vision guided embryonic microinjection robot**" is acceptable for publication in GENETICS. Many thanks for submitting your research to the journal.

To Proceed to Production:

1. Format your article according to GENETICS style, as discussed at <https://academic.oup.com/genetics/pages/general-instructions>, and upload your final files at <https://genetics.msubmit.net>.
2. Your manuscript will be published as-is (unedited-as submitted, reviewed, and accepted) at the GENETICS website as an Advanced Access article and deposited into PubMed shortly after receipt of source files and the completed license to publish. Please notify sourcefiles@thegsajournals.org if you do not wish to publish your article via Advanced Access.
3. We invite you to submit an original color figure related to your paper for consideration as cover art. Please email your submission to the editorial office or upload it with your final files. You can submit a small-sized image for evaluation, and if selected, the final image must be a TIFF file 2513px wide by 3263px high (8.375 by 10.875 inches; resolution of 600ppi). Please avoid graphs and small type.

If you have any questions or encounter any problems while uploading your accepted manuscript files, please email the editorial office at sourcefiles@thegsajournals.org.

Sincerely,

Jill Wildonger
Associate Editor
GENETICS

Approved by:
Kate O'Connor-Giles
Senior Editor
GENETICS

note: Please add jnls.author.support@oup.com and genetics.oup@kwglobal.com (or the domains @oup.com and @kwglobal.com) to your email program's "safe senders" list. You will be contacted by both at various points during the production process.

Review comments (if applicable): Please double-check that the Data Availability Statement is included; not just the Code Availability Statement.